# Assist-Dermo: A Lightweight Separable Vision Transformer Model for Multiclass Skin Lesion Classification

**DOI:** 10.3390/diagnostics13152531

**Published:** 2023-07-29

**Authors:** Qaisar Abbas, Yassine Daadaa, Umer Rashid, Mostafa E. A. Ibrahim

**Affiliations:** 1College of Computer and Information Sciences, Imam Mohammad Ibn Saud Islamic University (IMSIU), Riyadh 11432, Saudi Arabia; ymdaadaa@imamu.edu.sa (Y.D.); meibrahim@imamu.edu.sa (M.E.A.I.); 2Department of Computer Science, Quaid-i-Azam University, Islamabad 44000, Pakistan; umerrashid@qau.edu.pk; 3Department of Electrical Engineering, Benha Faculty of Engineering, Benha University, Qalubia, Benha 13518, Egypt

**Keywords:** skin cancer, pigmented skin lesions, dermoscopy, classification, deep learning, vision transformers, SqueezeNet, depthwise separable CNN

## Abstract

A dermatologist-like automatic classification system is developed in this paper to recognize nine different classes of pigmented skin lesions (PSLs), using a separable vision transformer (SVT) technique to assist clinical experts in early skin cancer detection. In the past, researchers have developed a few systems to recognize nine classes of PSLs. However, they often require enormous computations to achieve high performance, which is burdensome to deploy on resource-constrained devices. In this paper, a new approach to designing SVT architecture is developed based on SqueezeNet and depthwise separable CNN models. The primary goal is to find a deep learning architecture with few parameters that has comparable accuracy to state-of-the-art (SOTA) architectures. This paper modifies the SqueezeNet design for improved runtime performance by utilizing depthwise separable convolutions rather than simple conventional units. To develop this Assist-Dermo system, a data augmentation technique is applied to control the PSL imbalance problem. Next, a pre-processing step is integrated to select the most dominant region and then enhance the lesion patterns in a perceptual-oriented color space. Afterwards, the Assist-Dermo system is designed to improve efficacy and performance with several layers and multiple filter sizes but fewer filters and parameters. For the training and evaluation of Assist-Dermo models, a set of PSL images is collected from different online data sources such as Ph2, ISBI-2017, HAM10000, and ISIC to recognize nine classes of PSLs. On the chosen dataset, it achieves an accuracy (ACC) of 95.6%, a sensitivity (SE) of 96.7%, a specificity (SP) of 95%, and an area under the curve (AUC) of 0.95. The experimental results show that the suggested Assist-Dermo technique outperformed SOTA algorithms when recognizing nine classes of PSLs. The Assist-Dermo system performed better than other competitive systems and can support dermatologists in the diagnosis of a wide variety of PSLs through dermoscopy. The Assist-Dermo model code is freely available on GitHub for the scientific community.

## 1. Introduction

Skin cancer is becoming more widespread in the Western world, with significant ramifications for both general skincare and the availability of dermatological treatments. Day after day, about 99,780 individuals in the United States are identified with melanoma or skin cancer. Among them, two or more are likely to die per hour. Skin cancer affects more individuals in the United States each year than all other cancers combined [1]. Europe accounts for 9% of the global population yet bears 25% of the worldwide cancer cases. If tumors are recognized and treated early, cancer mortality can be considerably decreased. Thus, it is crucial to devote research resources to implementing systems for primary cancer recognition. Late-stage melanomas have a poor prognosis, like other malignancies. However, an early-identified melanoma can be treated by a simple resection. Early diagnosis and treatment can preserve the lives of far more than 95% of patients. These numbers are why so much work has been invested in evolving novel imaging approaches that improve the presentation of skin lesions and automated methods to detect melanomas. Human interpreting is laborious and susceptible to faults [2]. Artificial intelligence techniques offer great assistance in skin blemish identification.

Due to the weak contrast of skin carious lesions, the wide intra-class variance of skin cancers, and the high level of likeness among cancerous and non-cancerous lesions, the presence of numerous defects in the image makes automated pigmented skin lesions identification difficult. Often, there is a relatively weak contrast between healthy and affected skin. Additional factors comprise differences in skin color, skin abnormalities, the occurrence of defects (such as hairs, dark spots, foams, rulers, and date markers), uneven lighting and outer black circles, the lesion’s precise position, but more crucially, the lesion’s individual changes regarding color, roughness, form, size, and arrangement within the frame of the image. When developing robust lesion separation algorithms, researchers should consider these aspects. The majority of these factors can have a minimal impact on lesion segmentation with the proper preprocessing steps [3,4]. Nine different forms of pigmented skin lesions are represented visually in Figure 1.

Recently, researchers worldwide have successfully utilized deep learning for different visual tasks [5,6]. Moreover, deep learning in dermatology is mainly used for lesion-specific diagnostic processes, identifying suspicious lesions among several benign lesions, and monitoring lesion progression over time. Skin cancer categorization has benefited significantly from growing study attention since it is amenable to visual shape identification. Research articles demonstrate that DL-based dermoscopic classifiers can enhance skin cancer diagnosis or are better than human specialists [7].

### 1.1. Research Highlights

We used cross validation to test our model and the results showed that it worked much better than the methods already being used.This study proposes a new classification model (Assist-Dermo) to recognize multiclass PSLs.A new preprocessing step is integrated into the perceptual-oriented color space to enhance contrast and adjust the light.The proposed Assist-Dermo model is constructed with many layers and various filter sizes but fewer filters, and these parameters are selected by using lightweight SqueezeNet on a depthwise separable CNN.It is assessed using experimental findings from the many datasets that were gathered with regards to sensitivity, specificity, and other metrics.The Assist-Dermo can reduce overfitting because the dense connection better protects against the overfitting problem, especially when learning from small amounts of data.

### 1.2. Research Outline

The remainder of the article is divided into the following sections: a survey of the latest research studies regarding automatic recognition of skin tumors in Section 2. The applied methodology including employed datasets the developed DL architecture is illustrated in Section 3. The findings of the suggested skin lesions DL model along with analysis of these results are demonstrated in Section 4. Lastly, conclusions and recommendations for further research are offered in Section 5.

## 2. Review of Related Research

Classification of skin images using older methods [3,4] required arduous preprocessing, segmentation, and feature extraction procedures. Researchers from all over the world have recently been using deep learning to solve a variety of visual problems with success [5,6]. In order to exploit the merits of both machine learning (ML) and deep learning methods some researchers [8,9,10,11,12] presented an integration between outmoded ML approaches and DL algorithms for segmentation or categorization of skin cancer lesions.

Researchers of [8] described a technique for segmenting cutaneous melanoma that combines fuzzy k-means (FKM) clustering and faster region-based CNN (FRCNN). In order to attain a fixed-length feature vector, FRCNN was used. The FKM was employed to partition the potentially cancerous zone of skin to variable size segments and borders. Three common databases—ISBI-2016, ISIC-2017, and PH2—were used for performance evaluation. Regarding the ISIC-2016, ISIC-2017, and PH2 databases, their proposed technique attained an average accurateness of 95.40%, 93.1%, and 95.6%, respectively. In [9], the authors proposed a hybrid Inception adaptive neuro-fuzzy (ANF) model for classifying dermoscopic photos into different seven labels. All images were first preprocessed using hat filter and segmented with Grabcut technique. Then the Inception was used to automatically attain discriminative features, which were fed to the ANF classifier. Their method achieved an accuracy of 97.91%, specificity (SP) of 98.70%, and sensitivity (SE) of 93.40% on the ISIC-2018 dataset. Meanwhile, in [10], the authors suggested a hybrid ResNet-SVM framework for efficient binary classification of skin lesions. Furthermore, the issue of a shortage of annotated datasets was resolved via transfer learning (TL) together with data augmentation. Among the three employed datasets, ISIC-2017, HAM10000, and ISBI-2016, the ISIC-2017 dataset showed 99.19% accuracy for the suggested approach. Similarly, in [11], the authors classified three skin lesions using four dissimilar CNN-based architectures, specifically, AlexNet, VGG, ResNet-18, and ResNet-101 for feature extraction and SVM, RF, and MLP for classification. To provide a sole categorization for the input skin lacerations, several classification findings were combined. The results on ISIC 2017 showed an AUROC of 87.3% for PSL categorization and 95.5% regarding seborrheic keratosis categorization.

In [12], the authors suggested using a combination of CNN and local descriptor encoding techniques to classify skin lesions. Using ResNet50 and ResNet101, the lesion features were recovered from the pictures. Then, using the ResNet features that were extracted, a fisher vector (FV) was employed to create a general image representation. Finally, classification was performed using an SVM and a Chi-squared kernel. ISBI 2016 dataset was used to assess the efficacy. These findings demonstrated that their method outperformed other methods. Particle swarm optimization (PSO) was employed by [13] for segmenting skin lesions utilizing a variety of techniques, including the “Firefly Algorithm (FA), spiral research activity, probability distributions, crossover, and mutation”. K-means was applied to enhance lesion segmentation. The creation of CNN made use of the hybrid learning PSO (HLPSO). Melanoma and nevus lesions could be distinguished using the classification system. In [14], the authors developed a hybrid classifier for skin cancer employing both ECOC-SVM and DCNN. A collection of colored images of different skin cancers were gathered from the internet. For features extraction, a pre-trained AlexNet model was employed. Then, ECOC-SVM was used for skin cancer differentiation. More than 3750 images comprised of four skin cancer classes were employed for assessing efficacy. Their findings exhibited an average accurateness of 93.35%.

However, the authors in [15] provided an “intelligent Region of Interest (ROI)” algorithm to distinguish tumors from nevus malignancies with the aid of a transfer learning technique. A more effective variation of the k-mean technique was utilized to take out the ROIs from photos facilitating the recognition of distinctive traits, since only photos including malignancy cells were used for training. Furthermore, transfer learning along with data augmentation were utilized for extraction of ROI from the “DermIS” and “DermQuest” databases. The accuracy values from the suggested system for DermIS and DermQuest were 97.9% and 97.4%, respectively. Additionally, a CNN architecture that was trained utilizing ROI pigmented skin lesions was proposed by [16]. An AUC of 0.96 was achieved by their method. In [17], the authors presented a CNN-based six-class skin lesion classification technique. The authors took out 5846 medical photos of PSL from 3551 cases. A training dataset of 4732 images, 2885 patients with bounding box annotations and a test set of 666 images from 666 cases. A FRCNN developed architecture attained an accuracy of 86.2% while the derma experts showed an accuracy of 79.5% on the test set.

To accurately identify the greatest architectures for binary classification of skin cancer in more than 24,000 images, Ref. [18] applied three CNN models (Inception-V3, VGG19, and ResNet). The ISIC-19 and ISIC-2020 datasets were utilized to assess the classification accuracy. The Inception-V3 outperformed the other tested architectures with an accuracy of 86.90%. In [19], a DL architecture based on cloud infrastructure was utilized to implement different models for accurate skin cancer identification. SqueezeNet, DenseNet, and Inception-v3 pretrained models had ROC AUCs around 0.99, surpassing the ResNet model.

In order to increase performance and efficiency, Refs. [20,21] implemented a CNN model with many layers, various filter sizes, and a lower number of filters and settings for skin cancer categorization. For efficacy evaluation, Ref. [20] utilized the ISIC2017, ISIC2018, and ISIC2019 sets. Their method helps derma experts to categorize skin lesions, since it achieved 94% precision, 93% SE, 91% SP, and 0.964 AUC in ISIC2017 for PSL identification and categorization. While in [21], the authors used an idScore dataset that mixed dermoscopic images with clinical and histological information for classifying atypical moles. They achieved an AUC of 90.3%, SE of 86.5%, and SP of 73.6%. In [22], they compared the performance of CNN-based binary melanoma classifier trained with dermoscopic images to CNN as input data joint with an ANN for patients’ metadata. The ISIC-2019 dataset was used for assessing the performance of both models. The CNN+ANN model achieved an accuracy of 92.34%, thus outperforming the CNN model, which only attained 73.69%. However, the authors did not discuss the low accuracy results of their CNN model compared to the most recent research.

For three well-known architectures such as Inception-V3, VGG16, and VGG19, the authors in [23] created models using the transfer learning insight. The ISIC-17 dataset was used for training these models, which includes 604 test photos and 2487 test set of seven dissimilar classes of PSL. Their results showed accuracy of 74% for Inception-V3, 77% for VGG16, and 76% for VGG19. The authors of [24] described the development of a computerized skin lesion categorization algorithm employing dermoscopic photos and CNN as the base classifier. The model’s highest level of accuracy was 88.6%.

A hybrid-CNN [25] was developed based on three different feature extracting modules. These steps were combined to produce lesion feature vectors with better depth. Their system achieved an AUC of 0.96, 0.95, and 0.97, when tested against the ISIC2016, ISIC2017, and ISIC2018 datasets, respectively. In [26], the authors presented the classification of four different forms of skin cancer using the SCDNet, a vgg16-based framework. A classification accuracy of 96.91% was attained by SCDNet. Additionally, the proposed method’s accuracy is contrasted with that of four cutting-edge pre-trained classifiers.

For precise skin lesion subdivision, a “fully convolutional residual network (FCRN)” was built in [27]. Experimental results showed the suggested architecture had improved significantly in terms of performance, placing it first among 25 teams for classification and second among 28 teams for segmentation. The authors in [28] employed a set of data including 129,450 medical pictures for CNN training. The most common malignancies are represented in the first case, while the worst skin cancer is shown in the second case. In both missions, CNN performed as well as all tested specialists, demonstrating that AI is proficient of categorizing skin malignancies with a level of efficacy analogous to derma experts. Using a unique regularization technique, Ref. [29] proposed a different forecasting scheme which divides skin lacerations into “benign or malignant” lesions. Their new model achieved a 97.49% average rightness evaluated on the ISIC-2018. However, their new regularization method is not usable for feature extraction. In addition, choosing the appropriate parameters is a tedious and time-consuming process.

The authors of [30] reported that the average accuracy attained by derma experts for skin cancer identification is between 62% to 80% and they proposed the use of four ensemble and five CNN models for multiple class skin lesions discrimination. The HAM10000 dataset was utilized to assess the performance after applying preprocessing. Their results indicated a 93.2% maximum accuracy for ResNeXt101 model while ensemble model achieved 92.83% accuracy. Meanwhile, Ref. [31] separated photos into benign and cancerous categories. Their models were trained and tested using the open ISIC2020 database. According to the ISIC-2020 database, melanoma is categorized as cancerous. The effectiveness of three pre-trained architectures was then reported with a classification accuracy of 98.39%. Deep learning-based methods are employed by a popular yearly competition created by the “International Skin Imaging Collaboration Project (ISIC)” [32,33].

The authors in [34] used transfer learning and a pre-trained deep learning architecture to classify skin lesions. Through substituting the last layer with a softmax aiming to divide lesions into three classes, “transfer learning” is applied to AlexNet as well as soft-adjustment and data enlargement. Utilizing the ph2 database, their proposed architecture was trained and tested. The efficacy of their architecture obtained values of 98.61% accurateness, 98.33% sensitivity, and 98.93% specificity. Whereas, in [35], the authors described the creation of an ensemble of deep CNNs to further improve the efficiency of each CNN while identifying dermoscopy photographs into three cancerous levels: “melanoma, nevus, and seborrheic keratosis”. They were unable to receive sufficient training on an appropriate quantity of annotated photographs. They examine many fusion-based aggregating techniques and choose the most effective one for this issue. Regarding the three categories categorization mission, the average AUROC was 0.891.

The authors in [36] suggested an architecture based on weighted mean ensemble learning to categorize seven different kinds of skin infections. As the foundation of the ensemble, they used five DL architectures: ResNeXt, SeResNeXt, ResNet, Xception, and DenseNet. About 18,730 dermoscopy photographs from the HAM10000 and ISIC-2019 datasets at the same time with class balance, noise elimination, and data enlargement techniques were utilized for the training and validation of the evaluated models. The weighted mean ensemble scored a 94% recall rate, whereas the basic mean ensemble model attained 93%. The impact was revealed via the grid search technique. In order to train a DCNN, authors of [37] employed 4867 clinical photos from 1842 patients who had been diagnosed with skin tumors as a dataset. The DCNN model achieved an accuracy of malignant/benign classification of 92.4%. In [38], the authors introduced a bilinear CNN strategy that made use of transfer learning, a soft-adjustment step, and data augmentation to enhance classification performance while lowering the computing cost. Over the HAM10000 dataset, several imitations were run. According to their results, a bilinear method using the ResNet50 and VGG16 models improved accurateness by 2.7% above the up-to-date architectures. For example, the suggested method needed 238.6 min to train and averaged 93.21% accuracy. Whereas in [39], the authors proposed a DCNN (GoogLeNet Inception-v3) architecture to categorize derma-microscopy images into seven categories comprising the merit of binary support, showing that these categories can be unified into only normal/cancerous.

Also, in [40], the authors trained a DL architecture with 220,680 photographs of 174 illnesses and assessed it. Their system provided multi-class categorization among 134 illnesses, provided accurate malignancy prediction, recommended initial treatment alternatives, and enhanced the performance of medical experts. AUC for detecting malignancy was 0.928 and 0.937. In [41], the authors matched deep learning’s efficiency to that of derma specialists in categorizing histopathologic melanoma photos. For categorization of histological melanoma photos, CNN moderately outperformed eleven histopathologists despite having access to less image data, obtaining a mean sensitivity of 76%, specificity of 60%, and accuracy of 68%. Table 1 presented a comparison of deep learning approaches for classifying skin cancer.

## 3. Materials and Methods

Since skin melanoma ranks among the highly common cancers worldwide, precise and non-invasive diagnosis based on dermoscopic images has become critical and promising. This paper introduces the Assist-Dermo system for classifying nine classes of PSLs through an advanced deep learning approach by integrating preprocessing and data augmentation techniques. The overall architecture of the Assist-Dermo system is presented in Figure 2. The classification model is trained on the selected dataset. An enhanced lightweight version of SqueezeNet architecture that adopted a depthwise-separable CNN model was developed. The key aim of this article is to identify nine malignance types using images of skin lesions such as “actinic keratosis (AK), basal cell carcinoma (BCC), dermatofibroma (DF), melanoma (MEL), nevus (NV), pigmented benign keratosis (PBK), seborrheic keratosis (SK), squamous cell carcinoma (SCC), vascular lesion (VASL)”. The proposed Assist-Dermo was tested on the 24,000 images dataset. This section illustrates the different stages of the proposed Assist-Dermo PSLs classification system.

### 3.1. Acquisition and Preparation of Dataset

This section explores the recently utilized datasets for skin cancer classification, focusing on multiclass. The International Skin Imaging Collaboration (ISIC) offered various skin lesion datasets starting in 2016. These datasets were collected from a fair trial of affected participants who had undergone skin malignancy testing at various organizations, utilizing a range of derma-microscopy procedures over all anatomical sites (apart from the mucous membrane and nails).

The HAM10000 (“Human Against Machine with 10,000 Training Images”) database, also known as the ISIC2018 dataset [32,33], consists of seven PSL classes. It was separated into training and test databases, each of which includes 10,015 and 1512 images, respectively. The ISIC-2019 (BCN 20000) [42] comprises 25,331 photos grouped into eight PSL classes: “Actinic keratosis (AK), basal cell carcinoma (BCC), dermatofibroma (DF), melanoma (MEL), nevus (NV), pigmented benign keratosis (PBK), seborrheic keratosis (SK), and squamous cell carcinoma (SCC)”. The ISIC2020 [43] database contains 33,126 dermoscopy pictures categorized into nine well-known PSL classes as well as an unidentified picture class. More than 2000 patients provided the dermoscopy photos for these eight PSLs. The same eight PSL classes of ISIC 2019 in addition to “vascular lesion (VASC)” make up the nine PSL groups of photos.

Table 2 shows these datasets, indicating the number of selected images as well as the number of PSL classes. Table 3 represents the initial collection of PSLs with respect to multiclass. Those PSL images were used by the data augmentation technique to eradicate the problem of data imbalance as mentioned in Table 4. The visual result of data augmentation techniques is shown in Figure 3.

We discovered that the number of photos included within the various classes of PSLs in the selected dataset varied significantly. For example, the NV class has many more samples than the other classes. Additionally, the DF and VASC categories have fewer samples. We require enough balanced data to successfully train a DL-based model. Data balancing is performed by purposefully generating the required samples to prevent biased sampling during the DL model’s training. In addition, the unbalanced data may cause the model training to continue favoring classes with many examples. Therefore, to balance our dataset we used data augmentation, which comprises nine classes to increase our dataset by more than 24,000 and balance it for each class. Because affine transformations like rotation and shearing proved to have a detrimental effect on performance, we decided to include this type of augmentation in our investigation. Additional augmentation processes include adding blur with a probability of 25%, adding random Gaussian noise, altering brightness and contrast, and randomly flipping an image horizontally. During the training process, these data augmentation techniques were used. Moreover, in the selected datasets, there were different sizes of the 24-bit RGB PSL images ranged from (512 × 718) to (2848 × 4288). To address the variable-sized pictures issue, all dermoscopic images were reduced to a consistent size (512 × 512 × 3) by an image resize technique.

### 3.2. Preprocessing

Color space conversion is a crucial step that even experts in medicine use to separate non-melanoma skin lesions from healthy skin to simplify the segmentation process. The chosen color space must therefore be compatible with how people perceive color. Additionally, it was discovered that the selection of color space had a significant impact on the final image classification stage. Dermoscopy images have been used in the literature in the “RGB (Red, Green, Blue), YUV, HSV (Hue, Saturation, Value), CIE L*a*b*, and CIE L*u*v*” color spaces. Although the color spaces “RGB, YCbCr, and YUV” are frequently used in raw data and coding standards, human perception is incompatible with them. CIE color standards, in contrast, are computationally more complex but perceptually more homogeneous. The color appearance model CIECAM02 [20] offered the best features of previous color models. “Brightness Q, lightness J, colorfulness M, chroma C, saturation s, and hue h” are the six dimensions of color appearance included in the CIECAM02 color form scheme.

For picture segmentation, it has been demonstrated that the CIECAM02 (JCh) color space performs better than even RGB [28] and HSV [25] and is capable of reproducing human visual experience in colors. Therefore, both the preprocessing and skin lesion segmentation processes in this research used the JCh color space. The normalized ranges of the three planes are determined and are defined as (h: [0, 360]/15, C: [0, 1]/0.43, and J: [0, 1]/0.43) in the JCh color space transition step. To emphasize the significance of hue and reduce computational expenses, the 15 and 0.43 values are derived by empirical investigation. J*C*h* stands for the upgraded CIECAM02 (JCh) components. The “J*C*h*” perceptually-based color standard is developed.

To enhance the contrast and adjust the light illumination, we employed a technique developed in [46] but in perceptual-oriented (JCh) color space. We have selected this technique because it supports both local and global contrast of pixels without disturbing lesions patterns. The local improvement in PSL images is then created using an h-plane of JCh color space by maintaining the local contrast. To produce a finding that reflects a compromise between global and local contrasts, a contrast-brightness-based fusion algorithm is finally used. This technique enhances the visual quality while maintaining the authenticity of the image. The three-color channels are sometimes treated separately in color picture enhancement techniques, which alter the hue of the source photos. Hue preservation is a crucial strategy for color enhancement algorithms because these methods frequently produce unnatural-looking photos. The concept of hue in the JCh scheme was employed in our technique. Equation (1) determines the hue of an individual RGB colored point, *p*, with r, g, and b values:(1)$$h\left(p\right)=\left\{\begin{array}{c} 0\;\;\;\;\;\;\;\;\;\;\;\;\;if\;r=g=b\\ \theta \;\;\;\;\;\;\;\;\;\;\;\;\;\;\;\;\;\;\;\;\;\;if\;b\leq g\\ 2\pi -\theta \;\;\;\;\;\;\;\;\;\;\;if\;b>g \end{array}\right\}$$
where, the θ parameter is calculated as:(2)$$\theta =arcos\frac{1/2(\left(r-g\right)+\left(r-b\right))}{({\left(r-g\right)}^{2}+\left(r-b\right)\left(g-b\right){)}^{1/2}}$$

Both the single pixel p1 and the other single pixel p2 have values of (r1, g1, b1) that are [0, L − 1]^3^. If and only if there are a(.) values R such that (r1, g1, b1) = a (r2, g2, b2) +d3 = T, where d3 = (1, 1, 1)^T^, L is the range of pixel values, and for 8-bit pictures, L = 256, then 3 have the same hue. The claim is a fundamental principle of hue preservation. The analysis of hue-preserving-based enhancement techniques shows that these techniques are superior to the conventional channel-by-channel improvements or techniques that just improve the intensity channel. In our approach for global contrast improvement, the range of pixel values of a colored photo Xc is first translated to the entire range [0, 1].
(3)$$X_{c}^{\prime }=(J-1)\frac{X_{c}-X_{min}}{X_{max}-X_{min}}$$
where L = 256 for 8-bit photos, and Xmin and Xmax are the image’s smallest and highest intensity values for its three-color channels, respectively. Using Equation (4), the rigid color photo Xc is transformed into the associated intensity photo J.
(4)$$J=0.299\times X_{c}^{\prime }+0.587\times X_{G}^{\prime }+0.114\times X_{B}^{\prime }$$
where the three-color channels of the stretched color image X are XR, XG, and XB. To obtain the appropriate increased intensity picture GI, a new contrast improvement algorithm is then employed for J. Finally, we create the final improved color image Gc using the hue preservation enhancement framework.
(5)$$G_{c}\left(k\right)=\left\{\begin{array}{l} X_{c}^{\prime }\left(k\right)\frac{G_{I}\left(k\right)}{I\left(k\right)}\;\;\;\;\;\;\;\;\;\;\;\;\;\;\;\;\;\;\;\;\;\;\;\;\;\;\;\;\;\;\;\;\;\;\;\;\;\;\;\;\;\;\;\;\;\;\;\;\;\;\;\;\;\;\;\;\;\;\;\;if\;\frac{G_{I}\left(k\right)}{J\left(k\right)}\leq 1\\ \frac{J-1-G_{I}\left(k\right)}{J-1-I(k)}\;(X_{c}^{\prime }\left(k\right)-J\left(k\right)+G_{I}\left(k\right))\;\;\;\;\;\;\;\;\;\;\;\;\;\;\;\;\;\;\;\;\;if\;\frac{G_{I}\left(k\right)}{J(k)}>1 \end{array}\right\}$$
where k is the index of the pixels in each color channel, and c represents the corresponding R, G, and B color channel. With this equation, we have examined and demonstrated the effectiveness of color preservation. Figure 4 shows the contrast enhancement results in a perceptual-oriented color space. 

### 3.3. Architecture of SqueezeNet-Light

The primary goal is to find a CNN architecture with few parameters that has competitive accuracy. For improved runtime performance, this paper modifies the SqueezeNet design into Squeeze-Light by utilizing depthwise separable convolutions (SepConv) rather than simple conventional ones. Overall steps of this architecture are mentioned in Algorithm 1. To categorize PSLs into nine types, our improved SqueezeNet-Light classifier was used. The structure of the Squeeze-Light is described in Figure 5a.

The fire module is one of SqueezeNet’s main components. The fire module uses several strategies, including reducing the quantity of 3 × 3 filters and substituting 1 × 1 filters. These two methods have both been used to reduce the number of variables. The expand layer comes after the squeeze layer, and it subjects the output of the squeeze layer to two distinct convolution procedures using 1 × 1 and 3 × 3 kernels. Like the squeezing layer, each convolution operation is preceded by batch normalization and followed by the GELU procedure. In our work, the 3 × 3 convolutions in the extend layer E3 are replaced with depthwise separable convolutions (SepConv), which considerably compress the fire module of SqueezeNet. Section 3.3.2 provides a detailed explanation of depthwise separable convolutions. This smaller fire module will thereafter be figured out as a “spark” module (Figure 4). The number of input channels is shown by C in the diagram. The number of the squeeze layer’s channels is S. The 1 × 1 and 3 × 3 extension layers’ channel counts are E1 and E3, respectively. The number of channels in the output depends only on the expansion layers.

We write the batch normalization and depthwise separable convolution layers with the abbreviations BN and SepConv, respectively (Figure 4a). In addition, SepFire denotes the substitution of SepConv for Conv in the fire layer, while SepFire + BN denotes the addition of the BN layer after the SepConv layer in the Fire module (Figure 5b).
**Algorithm 1**: ShuffleNet-Light Architecture for Features Extraction and Classification of PSLs***Input:*** *Input Tensor (*X*), 2-D of (256 × 256 × 3) PSLs training dataset.****Output:*** *Obtained and Classified feature map*x=(x1,x2,……,xn)*augmented 2-D image****Main Process:****Step 1. Define number of stages = 4**Step 2. Iterate for Each Stage**(a)* *“Depthwise-CNN* *is applied to tensor x by kernel size of (3 × 3), which includes a number of filters; branch normaliza-tion, the ReLU activation function, Pointwise-CNN by kernel size of (1 × 1), branch normalization, and the GELU” activation function is applied.**(b)* *“Pointwise-CNN* *is applied to tensor x by kernel size of (1 × 1), which includes a number of filters, branch normalization, ReLU activation function, Pointwise-CNN by kernel size of (1 × 1), branch normalization, GELU” activation function is applied.**Step 3. Fscale = Squeeze and Excitation (SE) block contains expansion (1 × 1 × 3) layers.**Step 4. Fcat(i) = concatenation (# features-maps)**Step 5. channel = shuffle (x)**[End Step 2]**Step 6. Model Construction**(a)* *Define* *global-average-pooling layer**(b)* *Define* *fully-connected (FC) Layer and applied GELU function.**Step 7. Afterward, the feature map*x=(x1,x2,……,xn)*generated, which is recognized by Softmax function.**Step 8. Test samples *yit *are predicted to the class label using the decision function of the below equation. *yit=∑t=0M−1ft(xi)

#### 3.3.1. SqueezeNet Neural Network

Recently, increasing the accuracy and number of network layers has been the primary research focus for deep CNNs. Existing deep CNNs demand a lot of hardware and have slow prediction performance because of their vast volume and number of layers. However, there are nine classification categories for the problem of classifying PSLs. Therefore, the network does not require a lot of layers and hyperparameters. As a result, a deep learning-based PSL identification system is particularly challenging to employ in actual scenarios. With fewer network layers, deep learning can produce superior classification accuracy, whereas using too many layers can overfit the model and produce low accuracy. A tiny DL network with high-level accuracy and a compact network design must be created as a result.

SqueezeNet is a sort of extremely small deep CNN that attains similar accuracy as AlexNet on ImageNet while using only one-fiftieth the number of parameters. Compared to standard deep learning networks, SqueezeNet is more widely deployable in a variety of scenarios, has faster training and testing speeds, and has minimal hardware configuration needs. SqueezeNet uses a more compact and effective framework, particularly in the following three areas:(1)To decrease network parameters, a smaller 1 × 1 convolution kernel is used in place of the 3 × 3 convolution (Conv) kernel;(2)The parameters and model volume are reduced using SqueezeNet’s developed fire layer;(3)A thinner (a smaller number of) pooling layer resulted; only three maxpooling levels and a global pooling-layer are present in SqueezeNet.

As a result, SqueezeNet has a greater output layer with improved classification accuracy and information retention. Figure 4 shows the SqueezeNet network structure diagram as well as the layout of the fire layer. Conv and Concat are the abbreviations for convolution and concatenation, respectively. Additionally, we changed the data’s input format to width (W), height (H), and channel (N). Figure 4a depicts the overall layout of SqueezeNet, which is made up of numerous fire layers. To prevent overfitting, the SqueezeNet network includes global pooling and dropout layers next to the Fire 9 unit before classifying with a Softmax classifier. Because there are few pooling layers in the architecture and they are positioned backward, a complete connection layer does not have to be added to the network, which greatly minimizes the volume of SqueezeNet and eases parameter modification. The fire layer’s precise layout is depicted in Figure 4b, and it utilizes a 1 × 1 convolution layer for linear projection. Then, to lower the model size, speed up the detection, and cap the number of input channels, the network is enlarged using a combination of 1 × 1 and 3 × 3 convolutions.

#### 3.3.2. Depthwise Separable Convolution

Even though SqueezeNet shrinks the size of the model and speeds up detection by fire layers, the efficacy of the conventional layer in SqueezeNet is constrained. Consequently, the model’s accuracy is not different from that of CNN. Hence, SqueezeNet network optimization is crucial. In this paper, aiming to enhance the SqueezeNet architecture, we substitute the conventional convolutions with a “depth-wise separable convolution”. The use of depthwise separable convolution allows for fewer training weight factors and fewer floating-point tasks, leading to a lighter, faster, and more accurate model [47].

Standard convolution operates by simultaneously handling the input channel and the convolution window to extract various characteristics in accordance with various convolution kernels. In depthwise separable convolution, two independent tasks are carried out. The first convolution in space task is carried out separately on every input channel while using a single 1-dimensional kernel to ensure an equal number of input and output channels (DConv). The channels computed in the preceding stage are then projected onto a new channel space using point-based convolution with a 1′1 kernel (PConv), as particularly presented in Figure 4c.

When biases are not considered, the number of input and output channels for a (N’N) standard conventional layer is C1 and C2, respectively. The standard convolutional layer needs the following number of parameters: C2 × C1 × N × N. However, SepConv simply needs C1 + N + C2 + C1 + 1 factors to produce an identical result. The SepConv layer’s parameters have been drastically cut back. Considering the larger convolution kernel, the SepConv parameters are considerably lower than those of regular convolutions. Hence, the impact on architecture optimization is greater. Batch normalization is employed after the SepConv or SepFire layers because it (1) increases the network’s training speed while stabilizing the distribution of input data for each layer, (2) simplifies the model parameter adjustment method to enhance the stability of architecture learning, and (3) lessens the issue of gradient disappearance, which, to some extent, regularizes the neural network.

The classical convolution is represented by Equation (6). While the depthwise separable convolution is mathematically represented by the Equations (7)–(9):(6)$${StdConv(\theta ,x)}_{\left(i,j\right)}=\sum_{h,w,c}^{H,W,C}\theta \left(h,w,c\right).x(i+h,\;j+w,c)$$
(7)$${DConv(\theta ,x)}_{\left(i,j\right)}=\sum_{h,w}^{H,W}\theta \left(h,w\right)\;*x(i+h,\;j+w)$$
(8)$${PConv(\theta ,x)}_{\left(i,j\right)}=\sum_{c}^{C}\theta_{c}\;.x(i,\;j,\;c)$$
(9)$${SepConv\left(\theta_{p},\theta_{d},x\right)}_{\left(i,j\right)}={PConv}_{\left(i,j\right)}(\theta_{p},{DConv}_{\left(i,j\right)}(\theta_{d},x))$$

#### 3.3.3. Network Parameter Optimization

The values of a function’s parameters to reach a minimum can be either approximated or algebraically derived through a closed-form solution. Iterative methods like gradient descent are frequently the only option because cost functions in machine learning depend on a vast number of variables, and there is almost never a feasible way to obtain a closed-form solution for the minimum. When utilizing gradient descent [47], the parameter values must first be initialized so that optimization can begin. The value of the cost function is then reduced by iteratively changing the parameter values. In each iteration, parameter values are modified in the opposite direction as the cost gradient, which lowers the cost.

Even though adaptive techniques generalize [48] less effectively than SGD for many models, like convolutional neural networks (CNNs), they are typically employed as the default method because of their stability in challenging situations, such as the SqueezeNet model. To simultaneously achieve three goals—training stability, good generalization (like SGD), and speedy convergence—we propose AdaBelief (like adaptive techniques). AdaBelief’s fundamental concept is to adjust the step size in accordance with one’s “belief” in the current gradient’s direction. We mistrust the present observation and move slowly if the observed gradient considerably deviates from the prediction; if the observed gradient is near the prediction, we trust it and move rapidly. This is achieved by considering the “exponential moving average (EMA)” of the noisy gradient as the forecast of the gradient at the following time step. Through extensive testing, we assess AdaBelief and prove that it beats rival methods for image classification and language modeling with brisk convergence and superb accuracy. The mathematical formulas are as follows:(10)$$Y_{predict}={model}_{W}(x)\times h$$
(11)$$w=w-\alpha \frac{\partial f(Y,Y_{predict})}{\partial w}$$

## 4. Experimental Results

In this section we illustrate the setup for all experiments along with a brief explanation for the employed performance metrices. In addition, we present the results of the conducted experiments as well as analysis for these results.

### 4.1. Experimental Setup

A processor from Intel, the Core-i7 7700HQ, with a frequency of 2.8 GHz and memory of 16 GB, along with a GPU from the NVIDIA GeForce series, the GTX-1050 Ti, with committed 4 GB of memory, are utilized as the experimentation hardware. We put the suggested deep Light-SqueezeNet approach for PSL multi-labelling into practice using the Python-based Keras environment, a high-level neural network API. The mini-batch size was fixed at 16, and the number of epochs was set to 40. In addition, we utilized adaptive belief (AdaBelief) to train the proposed model with a learning rate of 0.001, and the learning policy is set to “step” with a gamma of 0.5. Also, we employed the AdaBelief optimization technique with the following configurations: default settings of 0.9, 0.999, and 1 × 10^−8^ for the exponential decay, the moment estimates, and epsilon, as shown in Table 5.

The datasets include photos of variable sizes. Therefore, we resized the images to the customary size of 512 × 512. In experiments, the data are divided into two groups: the training set and the test set, and a 10-fold cross-validation test is applied. The test set is utilized for model evaluation and prediction. To ensure fair comparisons, the same settings are used for all models.

### 4.2. Model Evaluation Metrics

In this study, classifications into nine categories were made using a particular dataset. We evaluated the results through applying “confusion matrix”, accurateness, precision, recall, F1Score, and the “Matthews Correlation Coefficient (MCC)”. The confusion matrix lists various variations of anticipated and real class values. The “true positive (TP)” and “true negative (TN)” values represent, respectively, the properly categorized benign and malignant cases. The wrongly classified benign and malignant cases are indicated, respectively, by the “false positive (FP)” and “false negative (FN)” values. The model’s accuracy, recall, and F-measure were calculated using the macro-average method for each of the nine categorization classes. MCC is a popular metric that is applied even if the classes have drastically diverse sizes because it is a balanced measure. Following is how these metrics are calculated:(12)$$Accuracy\;\left(ACC\right)=\frac{TP+TN}{TP+TN+FP+FN}$$
(13)$$Precision\;\left(PR\right)=\frac{TP}{TP+FP}$$
(14)$$Recall\left(RC\right)=\frac{TP}{TP+FN}$$
(15)$$F1score=2\times \frac{PR\times RC}{PR+RC}$$
(16)$$MCC=\frac{TP\times TN-(FP\times FN)}{\left(TP+FP)(TP+FN)(TN+FP)(TN+FN\right)}$$

We carried out the trials utilizing several potent CNN models, namely VGG16, AlexNet, InceptionV3, GoogleNet, Xception, MobileNet, SqueezeNet, and SqueezeNet-Light techniques for recognizing multiclass scenarios. In addition, we have also performed SOTA comparisons. We utilized the validation and testing splits to assess the effectiveness of these procedures. The primary distinction between the two splits is that, while the testing data were derived from separate sources, the validation split was produced using the same sources as the training data. We review the two, three, five, seven, and nine-class scenarios. In all experiments, all performance measures are computed using a 10-fold cross-validation technique split by the prevalence of the diagnosis category. The means of the metrics are calculated and provided in the remaining sections of the study to evaluate the models’ efficacy.

### 4.3. Network Settings

In addition to accuracy, the loss function plays a crucial role in image categorization during the transfer learning (TL) training phase. An optimizer is applied to the weights and learning rate of the network. This step is performed to cut down on loss functions while training in the DL layers. In this paper, the weight and acceleration time inside the model layers of each pre-trained TL model were changed by applying various optimizers such as ADAM, SGD, Adadelta, AdaBelief, and RMSprop. The optimizer in DL algorithms manages weight and bias throughout the network fitting process. This is the main emphasis of this optimizer. We choose the best hyperparameters for the layer feature maps, filter size, activation function, pool size, dropout, and fine-tuning. Then, the optimization step was performed to attain the best parameters. This clarifies that each optimizer has a specific function in various deep-learning applications.

In contrast, AdaBelief optimization is used to determine the best set of hyperparameters to use when assessing the entire deep learning framework. A total of 12 hyperparameters are used in this work. We are trying to examine hyperparameters and fine-tuning layers, which include freezing the top or lower network layer. The automated hyperparameters and fine-tuning steps are going through various stages of evaluation to find the best set of hyperparameters and layers for the recognition of nine classes of PSLs. When using the AdaBelief method to evaluate the hyperparameter and fine tune the pre-trained layers, a default value for the hyperparameter and fine-tuning layers must be set. After completing automatic hyperparameter tuning and automated fine tuning against all of the TL models by using different optimizations, the second stage involves assessing the performance of different transfer learning (TL) models. The final section examines how well TL performs when some layers are frozen throughout the automated hyperparameter tweaking procedure. Each TL model was optimized using hyperparameters and fine-tuning steps.

### 4.4. Computational Cost

According to the complexity of computations, SOTA models and the suggested Light-SqueezeNet system were also compared. According to Table 6, the suggested DL architecture required a total processing time of about 184.5 s. Where the total processing times were 246.2, 230.1, 217.4, 211.8, 207.5, 195.7, and 193.4 s for the VGG16, AlexNet, InceptionV3, GoogleNet, Xception, MobileNet, and SqueezeNet, respectively. Accordingly, our suggested Light-SqueezeNet technique took less time to identify several PSL classes, which is essential in a setting where computational performance is crucial. This demonstrates how effective the suggested idea is in relation to the current paradigm.

It has been shown that SqueezeNet-Light may improve network detection performance and is better at finding nine classes of PSLs in images than traditional convolutional layers. Therefore, on several datasets, the suggested approach outperformed SOTA systems. The parameter size was reduced in part thanks to this separable transfer network. Table 7 compares the number of parameters in the convolutional layers of the VGG16, AlexNet, InceptionV3, GoogleNet, Xception, MobileNet, SqueezeNet, and the proposed SqueezeNet-Light CNN. The findings showed that the suggested model significantly reduced the number of parameters on the convoluted layer. According to the experiment’s findings, reducing parameters did not produce degenerate models but rather more network generalization. In conclusion, the suggested SqueezeNet-Light performs better than certain other conventional models, especially when using large datasets, and lays the framework for its usage in PSL analysis systems thanks to its faster running speed.

### 4.5. Visual Feature Representation

SqueezeNet-Light provides an efficient and effective model for the classification of multiple classes of PSLs on a vast dataset. The SqueezeNet-Light is used as an efficacious feature extractor, which is exhibited in Figure 6. Figure 6 shows a visual comparison of how the proposed (SqueezeNet-Light) model and the classic (SqueezeNet) model affect the feature reactions of PSLs. The images in the first left column are input PSL images, along with their feature reactions in the correct columns of Figure 6. It is noticeably observed that the patterns of PSL using the original SqueezeNet are unclear, including noise and vital details that are missing (Figure 6a). Although, in Figure 6b images, using our SqueezeNet, the PSL’s crucial texture details are easily noticed. Accordingly, our suggested model feature extractor effectively extracted the crucial textural information needed to distinguish distinct PSL lesions. As a result, our proposed model can handle the pattern classification problem of PSLs with a better response while ignoring artifacts like hairs.

### 4.6. Performance of Proposed System

The dropout value (the percentage of deleted layers at the training phase) and initial learning rate were selected at random. To avoid its unique and accidental nature and to decrease the influence of manual parameter modification, the network was taught to choose the best SqueezeNet architecture. The dropout rate ranged from 0.2 to 0.6. The SqueezeNet model contained batch normalization layers, allowing us to start training at a higher learning rate. As a result, their initial learning rates were between 102 and 104. Under batch sizes of 32, 64, and 128, the initial learning rate and dropout values are chosen at random, and the network is trained using the Adam optimization technique. For training under different training batches, each network contained 30 unique initial sequence parameters. The 10,000 training sessions were allowed for each network with combined parameters. The training was over before ten training sessions, when the verification set loss remained constant. Statistical analysis was used to calculate the accuracy (ACC), specificity (SP), sensitivity (SE), precision (PR), recall (RL), F1-score, and MCC values. These metrics were employed for the performance assessment of the proposed system and to compare it to formerly built, pre-trained transfer learning algorithms. The different experiments measured the accuracy in the convolutional layers of the VGG16, AlexNet, InceptionV3, GoogleNet, Xception, MobileNet, SqueezeNet, and SqueezeNet-Light models. This section contains an exhaustive overview of the many investigations conducted to assess the efficacy of the purported SqueezeNet-Light model. In addition, we used the area under the receiver operating curve (AUC) to show the effectiveness of the training and validation datasets with a 10-fold cross-validation test. Figure 7, Figure 8 and Figure 9 reveal the best plot loss, accuracy, AUC, and recall on the train and validation sets with data augmentation, run for 40 epochs, for the proposed SqueezeNet-Light model.

**Experiment 1:** This experiment showed a 10-fold cross-validation testing methodology to compare the outcomes in terms of the confusion matrix when the pre-processing step is used in different color spaces. Figure 10a–c shows the results of the proposed model with preprocessing in different color spaces like CIECAM, CIELab, and HSV. Figure 10d–f shows CIECAM, CIELab, and HSV color spaces without a preprocessing step to increase contrast and adjust brightness. As shown in these confusion matrices, the color space CIECAM is perfect for feature extraction and classification tasks.

**Experiment 2:** We used 10-fold cross-validation testing to compare the statistical results to different TL algorithms, such as VGG16, AlexNet, InceptionV3, GoogleNet, Xception, MobileNet, SqueezeNet, and SqueezeNet-Light models. Table 8, Table 9 and Table 10 show the classification results of the DL models that had already been trained based on different batch sizes like 16, 32, and 64. The performance of the developed SqueezeNet-Light system has been the same with all different batch sizes (16, 32, and 64). But the effect of batch size was observed in terms of the number of parameters and computational time. However, the classification results remain the same for other pre-trained TL algorithms. The developed approach yielded excellent results: SE of 94%, SP of 96%, ACC of 95.6%, PR of 94.12%, F1-score of 95.2, and MCC of 96.7, as well as a low training error (0.76) in identifying multiclass PSLs.

**Experiment 3:** In this experiment, the impact of various optimization techniques on classification results was examined. We have also utilized different optimizers to build an efficient SqueezeNet-Light model. Adaptive algorithms like Adam have a good convergence speed, while algorithms like stochastic gradient descent (SGD) have better generalization (better response to new data). To combine both optimizer qualities, AdaBelief was developed in the past as an optimizer to control the loss function. We believe that AdaBelief can take care of regions with “large gradient, small curvature” cases, while Adam does not handle them. In the same 10-fold of training data, Table 11 compares the optimizers with the weighted ones. The AdaBelief optimizer, with learning rate, weight decay, and momentum set at 1 × 10^−5^, 1 × 10^−8^, and 0.9, respectively, is compared to the different optimization methods. When training the model with the AdaBelief optimizer with momentum, the number of epochs is set to 40. This is because previous experiments have shown that the value of the loss function curve keeps going down after the 30th epoch and that convergence is not reached at the end of the 30th epoch. The epoch number is fixed to 40 using all optimizers for the sole purpose of this study to provide a fair comparison. Table 10 provides a summary of the numerical outcomes. When the AdaBelief optimizer is used, the SE value is seen to increase considerably to 94%. As a result, the experiments’ optimization method of choice is the AdaBelief optimizer.

**Experiment 4:** In this experiment, we have also evaluated different loss functions. The findings demonstrate that the weighted-cross entropy loss function can improve classification performance by correcting class imbalance. Cross entropy loss was 82.2, 97.8, 96.9, 76, 79, and 82.2, whereas weighted-cross entropy loss was provided at 94, 96, 95.6, 94.12, 95.2, and 96.7 for SE, SP, ACC, PR, F1-score, and MCC metrics, respectively. As a result, we have used the weighted-cross entropy loss function. Table 12 demonstrates those results.

**Experiment 5:** In experiment 5, we have also evaluated the proposed SqueezeNet-Light model in terms of computational cost based on different benchmarks. Compared to other CNN and TL-based architectures, the SqueezeNet-Light model is an effective method for classifying PSLs. The original SqueezeNet network has more parameters and FLOPs compared to the proposed SqueezeNet-Light model (shown in Table 13). As a result, the proposed enhanced architecture has fewer parameters and converges faster than its baselines. Table 12 shows that the SqueezeNet-Light model has FLOPs of 68.3 MFLOPs, a model Size of 9.3 MB, and a GPU Speed of 0.7 S. As a result, the SqueezeNet-Light model created a new and improved architecture with computational efficiency, as detailed in Section 3.

**Experiment 6:** In this experiment, we compared the execution of the proposed SqueezeNet-Light model on CPU, GPU, and TPU with respect to batch size and computational speed. Layer-by-layer examination of the CNN implementation on the CPU, TPU, and GPU is necessary in practice [43]. The SqueezeNet-Light network should be built with each job being a multiple instruction, single data (MISD) task to maximize its performance in TPU. Prioritizing the neural network’s tasks is necessary when building a network. In actuality, the GPU provides more programming simplicity and flexibility for small programs. GPUs are better suited to small batch quantities of data because of the execution pattern in wraps and scheduling on straightforward on-stream multi-processors. The GPU performs well for large datasets and network models by maximizing memory reuse. Fully linked neural networks have lesser weight reuse, which causes increased memory traffic as the model size increases. The GPU can be utilized for applications that require memory because of its memory bandwidth. When processing big neural networks, GPUs outperform CPUs due to their added parallelism capacity. For fully-linked neural networks, the GPU performs better than the CPU, while the TPU shines when dealing with large batch sizes.

Whereas in the case of TPU, the array structure has been utilized, which works better on the SqueezeNet-Light architecture with large batches to offer high throughput during training. Large batches of data are necessary to properly utilize the matrix multiply units in the systolic array of the TPU. As the batch size increases, the architecture speeds up. Due to the networks’ ability to reuse space for big batch sizes and intricate CNNs, TPU is the best. The performance of the proposed SqueezeNet-Light model’s CPU, TPU, and GPU benchmarks in terms of batch size is shown in Table 14.

These experiments showed that SqueezeNet-Light produces classification rates for nine classes of PSLs that are superior compared to those of standard models. The improved SqueezeNet-Light-based classification accuracy of PSLs was 95.6%. As a result, the use of SepConv instead of classical convolution layers in the SqueezeNet model provided the best architecture for developing a lightweight Assist-Dermo system. In this paper, the speed and the number of parameters of the model are the key performance measures in addition to classification accuracy.

### 4.7. Comparisons with SOTA

Comparisons were made between traditional models such as Salama-ResNet-SVM [10], Ashraf-FRCNN [15], Naeem-VGG16 [26], Hosny-AlexNet [34], Fujisawa-DCNN [37], Harangi-Inception [39], Hekler-CNN [41], SqueezeNet, and the proposed SqueezeNet-Light model. As the identification and classification models, the SqueezeNet-Light-based models with the highest verification set under various configurations were chosen. The training set based on 10-fold cross validation was used to train the deep learning models for comparison, and the verification set adjustment parameters were used to choose the best model. The performance of the classifier with regards to accuracy, recall rate, precision rate, and F1-score of several approaches that rely on the unbalanced data set was quantifiably explored by evaluating each model with the test set.

We calculated the typical test time (forward propagation time), the number of parameters, and the model size 1000 times for each image in the test set. Based on the detailed indicators, our aim is to choose the model with the best F1 score, accuracy, and detection speed. Table 14 provides information on the sensitivity, specificity, precision, accuracy, and F1-score of each model on the test set. The computational cost, total parameters, model sizes, and average prediction time for the 1000 test images are shown in Table 15. In this table, we demonstrate that SqueezeNet is faster at runtime but is more accurate thanks to the additional batch normalization layers. Although the performance is significantly slower after the batch normalizing layer is added, precision increases. SqueezeNet-Light contains an additional deep separable convolution layer in addition to the added batch normalizing layer. SqueezeNet-light outperforms SqueezeNet in terms of accuracy, precision rate, and F1 score, among other metrics. The pace of detection and classification has also been enhanced. Additionally, SqueezeNet-light has far fewer model parameters than SqueezeNet and AlexNet, which makes it more usable and feasible for implementation in a realistic context.

When compared to other SOTA models, SqueezeNet-light shows an accuracy that is higher than that of the original SqueezeNet and other SOTA TL-based CNN models. The proposed SqueezeNet-Light’s accuracy rating (95.6%) is higher than classical SqueezeNet (87.6%) and even Inception (85%). While it has a considerably shorter prediction time and a smaller model size than all other SOTA models. Those statistical results are mentioned in Table 15 and Figure 11. Additionally, the SqueezeNet-Light model reaches the model’s smallest volume. SqueezeNet-Light is therefore superior to other models in terms of accuracy, prediction speed, and model size for the detection of PSL lesions.

In this study, dermoscopic pictures are used to differentiate between various classes of pigmented skin lesions (PSLs) using a variety of datasets. A visual example of the results attained using our proposed SqueezeNet-Light classifier is displayed in Figure 12 to recognize nine classes of PSLs. The testing splits for other classes were also gathered from various sources. We assessed various pre-trained TL CNN architectures such as VGG16, AlexNet, InceptionV3, GoogleNet, Xception, MobileNet, and SqueezeNet techniques to discriminate among various skin cancer types in various scenarios. We employed data augmentation to avoid the issue that CNN architectures need a lot of labeled data to train on. The findings obtained demonstrated that our proposed model beats CNN architectures. In three-class, five-class, seven-class, and nine-class scenarios, our suggested SqueezeNet-Light architecture had an accuracy of 95.6% in the recognition of multiclass PLS classes. Computational performance of different SOTA architectures are mentioned in Table 16. According to this table, our proposed model achieved low computational burden as compared to other architectures.

## 5. Discussion

There is an urgent need to address this worldwide public health concern, since skin cancer incidence rates have been rising over the past few decades. Pretrained-based deep-learning CNNs are now being used for the classification of skin cancer due to their outstanding effectiveness in classifying medical images. Although numerous studies have been conducted in the past to classify skin cancer, none of them have been successful in expanding their research to include nine different types of skin cancer. In this study, we classified skin cancer more accurately than dermatologists and existing deep learning techniques. On the different datasets, the performance of the proposed SqueezeNet-Light is examined to identify which approach is most effective for classifying skin cancer. We conducted a significant study to identify the ideal configuration of hyperparameters. Comparing SqueezeNet-Light to earlier proposed deep learning models, the performance and computational efficiency have significantly improved. Therefore, we suggest using SqueezeNet-Light to classify skin cancer. We also concluded that deep learning models trained with the ideal hyperparameter setup cannot perform as well as other pre-trained CNN models. While these methods are frequently used to increase classification job accuracy, they also greatly increase the model’s architectural complexity and may not have a major impact on how well deep learning models perform when set up with the best hyperparameters.

The workload of dermatologists can be significantly reduced by creating an automated classification system for skin lesions, which will also lessen subjectivity and human error-related subjectivity in the classification process. There have been a few incidents of inappropriate therapy because of an inaccurate or delayed diagnosis. Misdiagnosis might occasionally result in a greater requirement for surgical intervention and a longer hospital stay because the effects of treatment can take time to manifest. The ability of dermatologists to perform skin examinations as diagnosticians has a significant impact on the detection of skin cancer. However, dermatologists with more than ten years of expertise scarcely top 80% memory, and those with three to five years of experience only reach 62% recall in the skin cancer screening. When competent dermatologists are hard to come by for unskilled practitioners in developing nations, this proposed method can become vital and more valuable. This SqueezeNet-Light model is effective at speeding up the automated classification process for skin lesions and can even be used to apply a class label to a brand-new skin lesion. While these results are reassuring and offer compelling evidence that the deep learning approach can help doctors and healthcare systems, more clinical data on topics like age, gender, race, and family history is still needed for further validation and advancement before the deep models can be evaluated in clinical practice.

Also, our method worked well in the three-class and five-class scenarios, with an overall identification accuracy (ACC) of 94.5%, sensitivity (SE) of 96.7%, specificity (SP) of 95%, and area under the receiver operating curve (AUC) of 0.95. We are attempting to gather additional hospital cases of skin cancer for future research. Moreover, based on the available dermoscopy scans, we intend to define additional skin cancer disease groups. The SqueezeNet-Light technique, on the other hand, aims to deploy more deep CNN architectures. Additionally, combining the advanced features of many architectures is an exciting tactic that can improve performance. A preprocessing step based on contrast enhancement in a perceptual-oriented color space is also needed to further improve the classification results. Additionally, skin lesions will be automatically separated in HSV color space using Grab-Cut with little human involvement. The ABCD (asymmetry, border irregularity, color, and dermoscopic patterns) rule to distinguish malignant melanoma from benign lesions will be automatically applied using image processing techniques. Different pretrained convolutional neural networks (CNNs), including InceptionV3 and MobileNet, are also evaluated to categorize skin lesions as benign or malignant. However, to categorize nine classes of PSLs, we have developed an effective and efficient classifier, Assist-Dermo.

Using a separable vision transformer (SVT) approach, a dermatologist-like automatic classification system is built to classify nine different kinds of pigmented skin lesions (PSLs) to aid clinical professionals in the early diagnosis of skin cancer. The authors have developed a few methods in the past to identify the nine PSL classes. However, they frequently require many calculations to attain good performance, making their deployment on devices with limited resources difficult. In this article, a novel SVT architectural design based on the SqueezeNet, and depthwise-separable CNN models is presented. The primary objective is to identify CNN architectures with minimal parameters and competitive precision. This study modifies the SqueezeNet architecture by employing depthwise separable convolutions rather than the basic conventional ones for enhanced runtime performance. An intuitive comprehension of CNN principles is provided by the display of the results. Plotting the feature maps obtained at successive levels, for instance, enables comparison of the modifications made to the features by the pooling, batch normalization, and activation layers. In the second scenario, the user can further investigate how the network’s performance is impacted by changes to the dataset and hyperparameters. Examples in this sense can include comparing performance when the learning rates of layers are changed or even frozen. Additionally, we think that making this technology more approachable for non-experts would strengthen the collaboration between dermatologists and computer scientists toward the joint effort of improving image-based medical diagnosis.

Table 17 demonstrates the limitations as well as the advantages of our SqueezeNet-Light DL architecture compared to SOTA architectures. Our SqueezeNet-Light architecture outperforms other DL methods in the number of classified PSL classes. Since it classifies the input PSL images into nine classes while others are ranged from two to seven classes without addressing the perceptual-oriented color space, the contrast is enhanced, and there is no control over class imbalance issues. Furthermore, the vast majority of them ignored classifier generalizability and trained and tested models on a single dataset. Moreover, the proposed SqueezeNet-Light model is a lightweight architecture since it has a tiny number of parameters and smaller memory requirements compared to other SOTA systems. Consequently, this leads to a faster processing and classification system.

In the future, we will also compare this proposed SqueezeNet-Light architecture with other recently developed lightweight [49,50] architectures to confirm the generalizability of the network. 

## 6. Conclusions

To aid clinical professionals in the early diagnosis of skin cancer, a derma expert-like automated categorization (Assist-Dermo) system is created in this work to classify several types of PSLs. Using a separable vision transformer (SVT) approach, a derma expert-like automated classification system is built to classify nine different kinds of pigmented skin lesions (PSLs) to aid clinical professionals in the early diagnosis of skin cancer. The authors have developed a few methods in the past to identify the nine PSL classes. However, they frequently require many calculations to attain good performance, making their deployment on devices with limited resources difficult. In this article, a novel SqueezeNet vision transformer (SVT) architectural design based on the SqueezeNet, and depthwise-separable CNN models is presented. The primary objective is to identify CNN architectures with minimal parameters and competitive precision. This study modifies the SqueezeNet architecture by employing depthwise separable convolutions rather than the basic conventional ones for enhanced runtime performance. For the development of this Assist-Dermo system, a data-augmentation approach was used to address the PSLs’ imbalance issue. Next, a pre-processing phase is used to choose the most dominant area, followed by the enhancement of lesion patterns in a color space geared toward perception. To increase efficacy and performance, the Assist-Dermo system is designed with many layers and numerous filter sizes but fewer filters and parameters. For the training and assessment of Assist-Dermo models, a collection of 24,000 photos of PSLs from online data sources such as Ph2, ISBI-2017, HAM10000, and ISIC is used to identify nine classes of PSLs. On the collected dataset, it obtained 95.6 accuracy, 94 sensitivity, 96% specificity, 94.12% precision, and a 95.2 F1-Score. The experimental findings demonstrate that the proposed Assist-Dermo approach outperforms existing algorithms in recognizing nine types of PSLs. The Assist-Dermo system performs better than current state-of-the-art methods and can help dermatologists diagnose a variety of PSLs via dermoscopy images. Also, the proposed Assist-Dermo method performed better than other SOTA methods at classifying different PSLs from dermoscopy images without accurate scarification in terms of speed of prediction and model size.

Computational efficiency shows that the proposed system can be easily deployed in an environment where is a requirement of resource-constrained devices such as mobile devices. However, it is important to note that the lightweight nature of the model should also be tested on the Internet of Things (IoT) devices, which makes it suitable for deployment on a wider scale for the accurate classification of pigmented skin lesions (PSLs). This point of view will be addressed in future works.

## Figures and Tables

**Figure 1 diagnostics-13-02531-f001:**
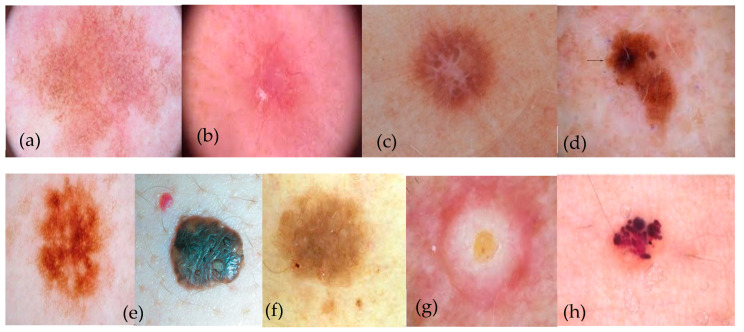
A visual example of nine types PSLs, where (**a**) actinic keratosis (AK), (**b**) “basal cell carcinoma” (BCC), (**c**) dermatofibroma (DF), (**d**) melanoma (MEL), (**e**) nevus (NV), (**f**) pigmented benign keratosis (PBK), (**g**) “seborrheic keratosis” (SK), and (**h**) “squamous cell carcinoma” (SCC).

**Figure 2 diagnostics-13-02531-f002:**
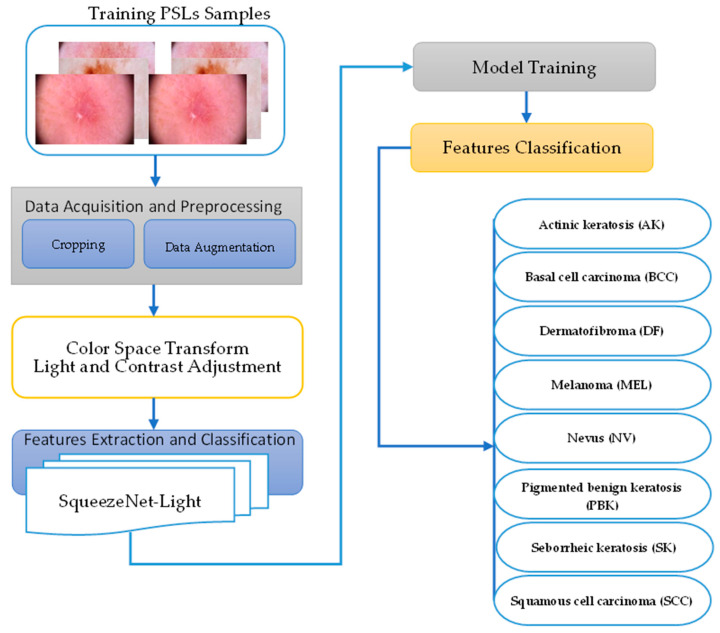
A methodical illustration of suggested Assist-Dermo system to recognize nine classes of pigmented skin lesions.

**Figure 3 diagnostics-13-02531-f003:**
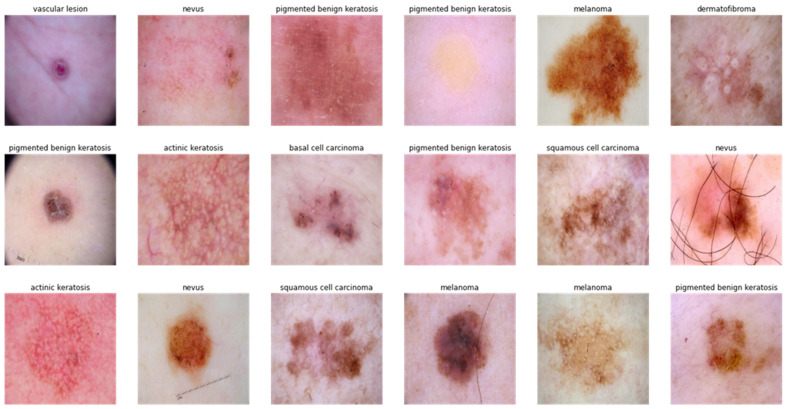
An example of data augmentation techniques applied on the selected datasets from the ISIC-2019, ISIC-2020, and HAM10000 sources in case of benign and malignant skin lesions.

**Figure 4 diagnostics-13-02531-f004:**
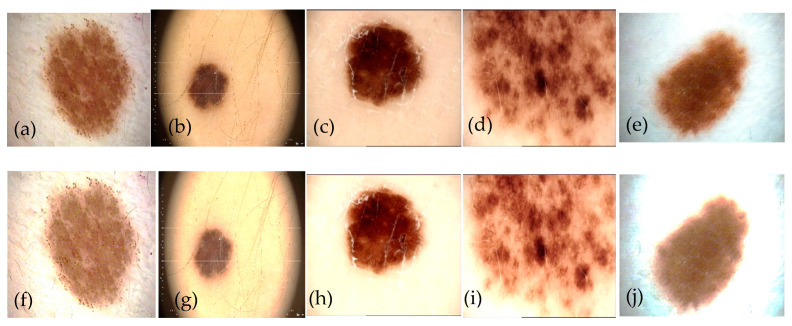
An example of preprocessing enhancement step to enhance the contrast of SCC lesion taken from Figure 1, where (**a**–**e**) represents the original PSLs, and (**f**–**j**) shows the corresponding contrast enhancement using nonlinear sigmoidal function.

**Figure 5 diagnostics-13-02531-f005:**
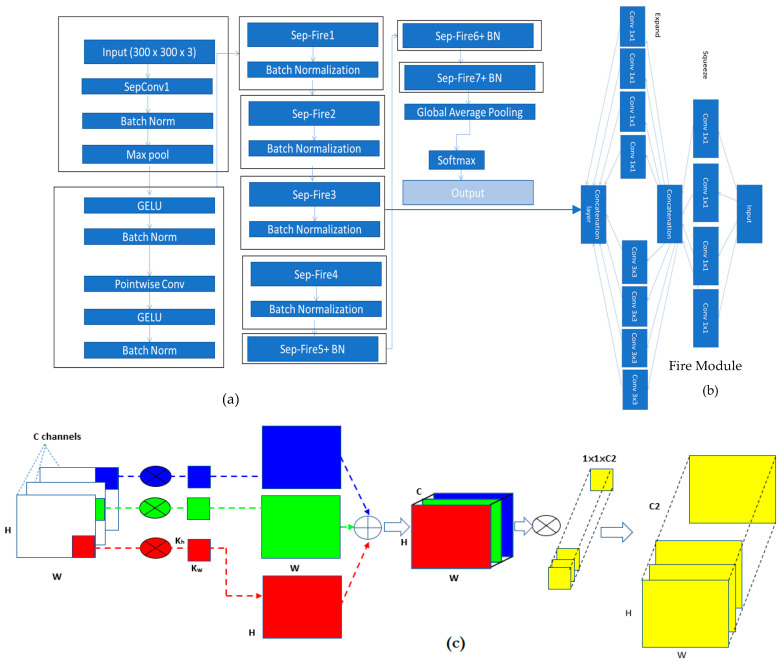
SqueezeNet-Light architecture (**a**) SqueezeNet-Light base structure, (**b**) fire module structure and (**c**) SepConv module basic structure.

**Figure 6 diagnostics-13-02531-f006:**
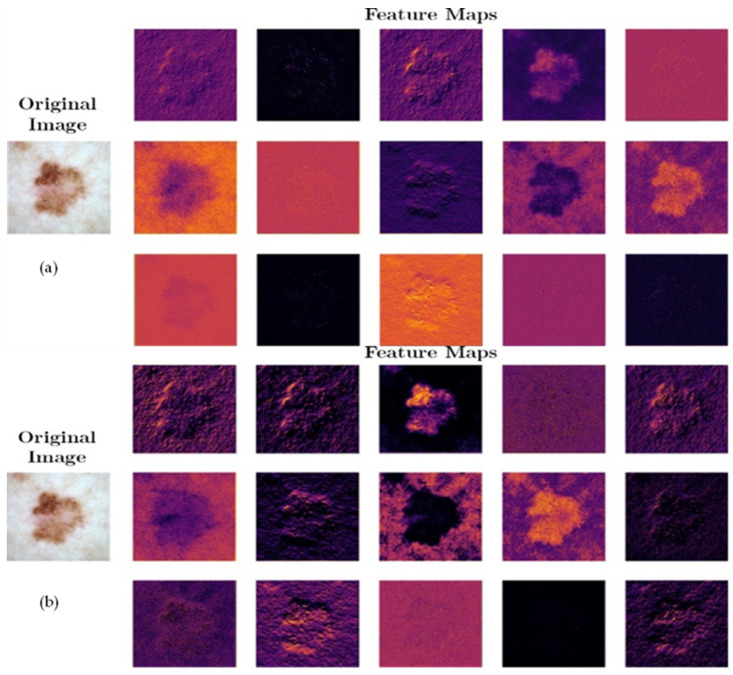
(**a**) Feature representations example of original SequeezeNet model using PSLs image and (**b**) feature representations example of our SequeezeNet-Light model using PSLs image.

**Figure 7 diagnostics-13-02531-f007:**
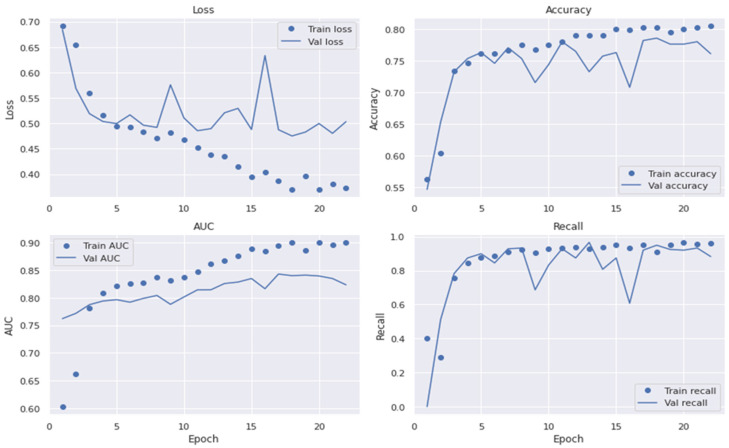
Plot of loss, accuracy, AUC, and recall on train/validation sets without data augmentations.

**Figure 8 diagnostics-13-02531-f008:**
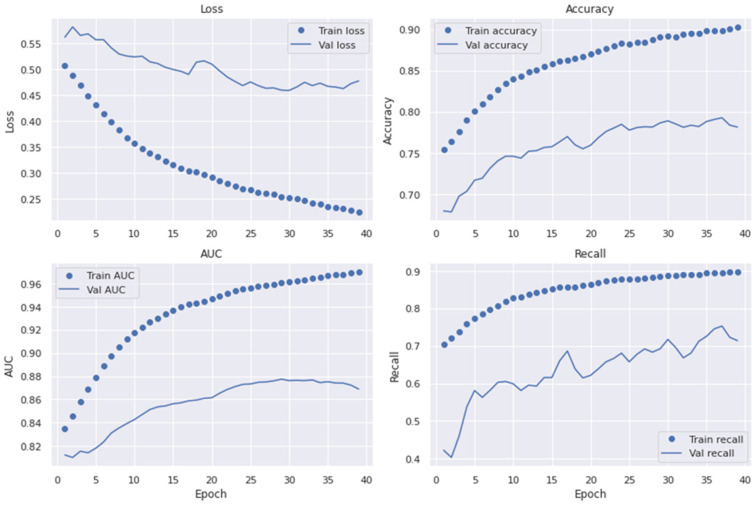
Plot loss, accuracy, AUC, and recall on the train and validation sets with data augmentation.

**Figure 9 diagnostics-13-02531-f009:**
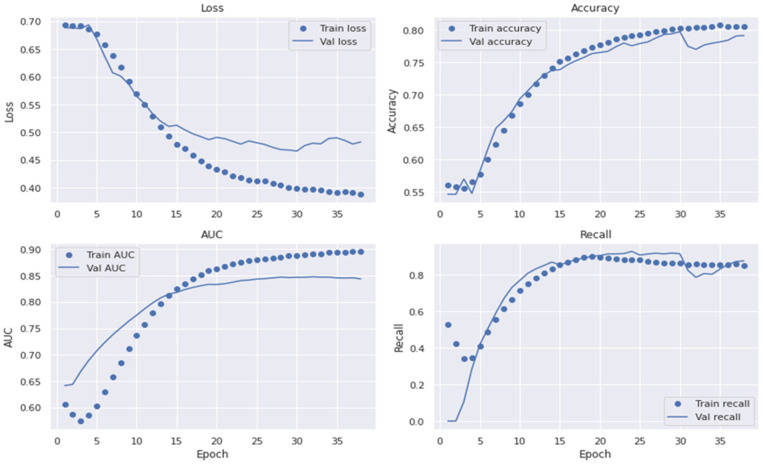
Plot loss, accuracy, AUC, and recall on train/validation sets with data augmentation and 40 epochs.

**Figure 10 diagnostics-13-02531-f010:**
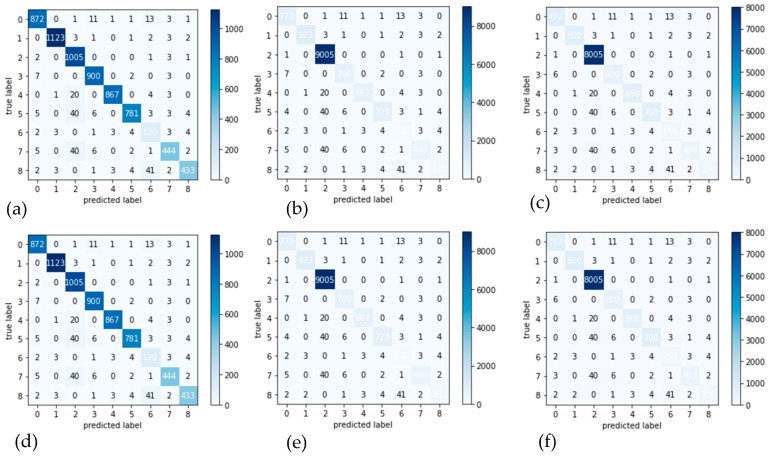
Confusion matrices: Sub-figures (**a**–**c**) show the result of proposed model with preprocessing in CIECAM, CIELab, and HSV color spaces. Sub-figures (**d**–**f**) show the result of proposed model without preprocessing step to enhance the contrast and adjust brightness, for the same color spaces.

**Figure 11 diagnostics-13-02531-f011:**
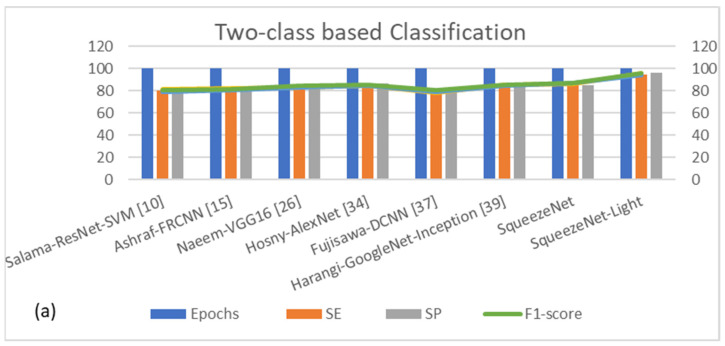
Performance of SOTA systems when compared to our Assist-Dermo system (**a**) binary classification (malignant and benign lesions), (**b**) represents five-classes’ classifications (AK, BCC, DF, Mel, NV), (**c**) illustrates seven-classes’ classifications (AK, BCC, DF, Mel, NV, PBK, SK), and (**d**) shows nine-classes’ classifications (AK, BCC, DF, Mel, NV, PBK, SK, SCC, Vasc).

**Figure 12 diagnostics-13-02531-f012:**
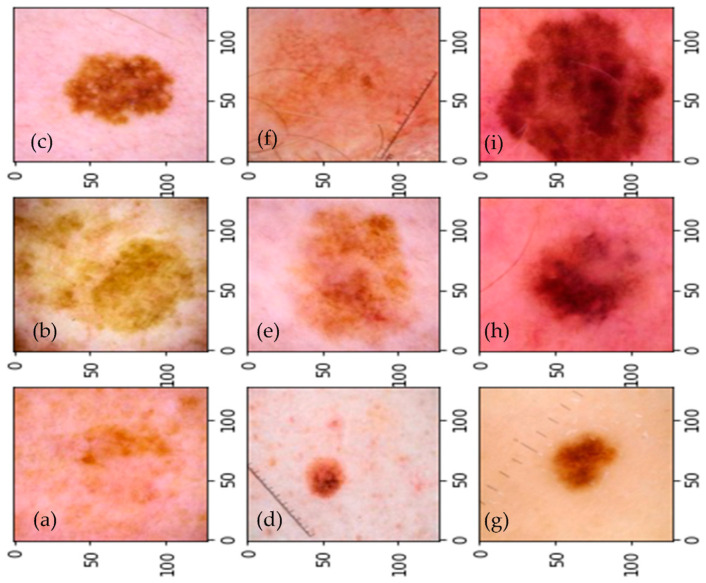
A visual example of proposed SqueezeNet-Light classification, where (**a**) actinic keratosis, (**b**) basal cell carcinoma, (**c**) dermatofibroma, (**d**) melanoma, (**e**) nevus, (**f**) pigmented benign keratosis, (**g**) seborrheic keratosis, (**h**) squamous cell carcinoma, and (**i**) vascular lesion.

**Table 1 diagnostics-13-02531-t001:** Comparison of DL-based approaches for classifying skin cancer.

Ref.	Diagnosis	Segment?	Classification	DL Model	Datasets	Result (%) **	Limitations
[9]	Proposed a hybrid Inception adaptive-neuro-fuzzy (ANF) model for discriminating dermoscopic photos into different seven labels.	Yes	7 classes	Inception-v4	ISIC-2018	ACC: 97.91% SE: 93.4% SP: 98.7%	Classes imbalance problem, evaluated on single dataset, classifies only seven classes, and computationally expensive.
[10]	Suggested a hybrid ResNet-SVM framework for efficient binary classification of skin lesions.	No	Binary	ResNet50, VGG-16 and SVM	ISIC 2017 ISBI 2016	ACC: 99.19%	Classes imbalance problem, binary classification only two classes, and computationally expensive.
[15]	Offered an augmented ROI -based CNN system to recognize and separate melanoma from nevus malignancy.	Yes	Binary	CNN + Transfer learning	DermIS, DermQuest	ACC: 97.9% ACC: 97.4%	Classes imbalance problem, evaluated on single dataset, classifies only two classes, and computationally expensive.
[17]	Introduced a six-class cutaneous lesions classification system based on CNN.	No	6 classes	Faster region-based CNN (FRCNN)	Private (5846 images)	ACC: Six-classes 86.2% Two-classes 91.5%	Image processing, handcrafted-based feature extraction approach, which limits the detection accuracy, 6 classes only, and computationally expensive
[19]	Suggested a hybrid ResNet-SVM framework for efficient binary classification of skin lesions.	No	Binary	SqueezeNet, DenseNet, inception v3 and ResNet	HAM10000	AUC: 0.997	Evaluated on one dataset, classifies only two classes, and computationally expensive.
[20]	Implemented a CNN model with many layers, various filter sizes, lower number of filters and settings for skin cancer categorization.	No	3 classes	DCNN	ISIC 2017 ISIC 2018 ISIC 2019	AUC: 0.964	Three classes of PSLs and reduced hyper-parameters, so computationally expensive.
[25]	Suggested a hybrid-CNN made up of three different feature extracting modules that were combined to produce lesion feature vectors with better depth.	Yes	7 classes	CNN	ISIC-2016, ISIC-2017, ISIC-2018	AUC: ISIC-2016: 0.96 ISIC-2017: 0.95 ISIC-2018: 0.97	Seven classes only and used only one limited dataset, classifier is not generalized.
[26]	Presented the classification of four different forms of skin cancer using the SCDNet, a vgg16-based framework.	No	4 classes	Vgg16	ISIC 2019	ACC: 96.91%	Four classes only, many hyper-parameters required, and tested on signal dataset, so classifier is not generalized.
[29]	Proposed a new regularization technique for CNN model to binary classify skin lesions.	No	Binary	CNN	ISIC-2018	ACC: 97.49%	Two classes (benign vs. malignant), required huge parameters, evaluate on single dataset, and computationally expensive.
[31]	Examined how well three of the best pretrained DL models classified skin cancer in binary form.	No	Binary	ResNet, VGG16, MobileNetV2	ISIC 2020	ACC: 98.39%	Two classes (benign vs. malignant), required huge parameters, evaluate on single dataset, and computationally expensive.
[34]	Used transfer learning and a pre-trained deep learning architecture to classify three skin lesions.	No	3 classes	AlexNet	ph2	Acc: 98.61% SE: 98.33% SP: 98.93%	Three classes only, required huge parameters, evaluate on single dataset, and computationally expensive.
[35]	Described the creation of an ensemble of deep CNNs to further improve the efficiency of each CNN while identifying dermoscopy photos into three categories	No	3 classes	GoogLeNet, AlexNet, ResNet, VGGNet	ISBI 2017	AUC: 0.891	Single dataset used, three classes of PSLs and reduced hyper-parameters, so computationally expensive.
[36]	Suggested an architecture based on weighted mean ensemble learning to categorize seven different kinds of skin infections	No	7 classes	ResNeXt, SeResNeXt, DenseNet, Xception, ResNet	HAM10000 ISIC 2019	ACC/recall: avg.: 87%/93% weight avg.: 88%/94%	Seven classes with two datasets used, three classes of PSLs and reduced hyper-parameters, so computationally expensive.
[37]	Employed 4867 clinical photos from 1842 patients who had been diagnosed with skin tumors as a dataset to train a DCNN architecture.	No	Binary	DCNN	Private (4867 images)	ACC: 92.4%	Two classes (benign vs. malignant), required huge parameters, evaluate on single dataset, and computationally expensive.
[38]	a bilinear CNN strategy that made use of transfer learning, a soft-adjustment step, and data augmentation to enhance classification performance while lowering the computing cost.	No	7 classes	ResNet50 and VGG16	HAM10000	ACC: 93.21%	Three classes only and used only one limited dataset, classifier is not generalized.
[39]	classified dermoscopy images into seven classes comprising the advantage of binary support.	No	7 classes	GoogLeNet and Inception-v3	ISIC 2018	BMA: 67.7%	Seven classes, required huge parameters, evaluate on single dataset, and computationally expensive.
[41]	Deep learning’s efficiency to that of derma specialists in categorizing histopathologic melanoma photos.	No	Binary	CNN	Private (695 lesions)	ACC: 68% SE: 76% SP: 60%	Two classes (benign vs. malignant), required huge parameters, evaluate on single dataset, and computationally expensive.

D dermoscopic, C clinical, H Histopathological WSIs. ** ACC: accuracy, SE: sensitivity, SP: Specificity, BMA: balanced multi-class accuracy, AUC: area under curve.

**Table 2 diagnostics-13-02531-t002:** Dermoscopic datasets brief description.

Dataset	Ref.	Images	Selected Images	Number of Classes *
HAM10000	[32,33]	10,015	10,015	7 (“AK, BCC, BKL, DF, NV, MEL, VASC”)
ISIC 2019	[42]	25,331	25,331	8 (“MEL, NV, BCC, AKIEC, BKL, DF, VASC, SCC”)
ISIC 2020	[43]	33,126	33,126	9 (“MEL, NV, BCC, AKIEC, BKL, DF, VASC, SCC, PBK”)
Ph2	[44]	200	200	2(NV, and Mel)
ISBI 2017	[45]	2750	2750	3 (NV, Mel, and SK)

* Nv: Nevus, Mel: melanoma, SK: seborrheic keratosis, BKL: “benign keratosis lesion”, DF: “dermatofibroma”, AK: actinic keratosis, BCC: “basal cell carcinoma”, VASC: “vascular lesion”, SCC: squamous cell carcinoma, and PBK: pigmented benign keratosis.

**Table 3 diagnostics-13-02531-t003:** Number of images in each class.

Classes *	No. of Images	Size
AK	700	(512,512,3)
BCC	3300	(512,512,3)
BKL	2600	(512,512,3)
DF	200	(512,512,3)
NV	12,000	(512,512,3)
MEL	4000	(512,512,3)
SCC	600	(512,512,3)
VASC	300	(512,512,3)
PBK	300	(512,512,3)
Total	24,000	(512,512,3)

* AK: Actinic keratosis, BCC: “basal cell carcinoma”, BKL: “benign keratosis lesion”, DF: “dermatofibroma”, NV: nevus, MEL: melanoma, SCC: squamous cell carcinoma, VASC: “vascular lesion”, and PBK: pigmented benign keratosis.

**Table 4 diagnostics-13-02531-t004:** Data augmentation techniques used to develop a light-dermo system.

Parameters	Angle	Brightness	Zoom	Shear	Mode	Horizontal	Vertical	Rescale	Noise
Values	30°	[0.9, 1.1]	0.1	0.1	Constant	Flip	Flip	1./255	0.45

**Table 5 diagnostics-13-02531-t005:** Hyper parameters configurations of the developed SqueezeNet-Light architecture.

Tensor Flow	GPU	Learning Rate	Optimizer	Number of Epoch	Batch size	Validation
2.9.1	GeForce GTX 1050 Ti	1 × 10^−3^	AdaBelief	40	16	10-fold

**Table 6 diagnostics-13-02531-t006:** Average processing time on a PSLs dataset by various DL algorithms.

Method	Preprocessing	Feature Extraction	Training	Prediction	Overall
VGG16	20.5 s	14.4 s	200.5 s	10.8 s	246.2 s
AlexNet	18.6 s	12.2 s	190.5 s	8.8 s	230.1 s
InceptionV3	16.3 s	14.8 s	178.5 s	7.8 s	217.4 s
GoogleNet	17.2 s	17.3 s	170.5 s	6.8 s	211.8 s
Xception	18.1 s	15.1 s	165.5 s	8.8 s	207.5 s
MobileNet	14.1 s	13.3 s	160.5 s	7.8 s	195.7 s
SqueezeNet	10.8 s	8.3 s	168.5 s	5.8 s	193.4 s
**Proposed**	**1.8 s**	**1.9 s**	**165.5 s**	**1.5 s**	**184.5 s**

**Table 7 diagnostics-13-02531-t007:** The number of parameters in the existing pretrained TL models compared to SqueezeNet-Light model.

Models	Image Size	Parameters	Validation Accuracy
VGG16	512 × 512 × 3 256 × 256 × 3 200 × 200 × 3	14,714,688 14,865,222 14,911,302	79
AlexNet	512 × 512 × 3 256 × 256 × 3 200 × 200 × 3	23,587,712 14,911,302	81.3
InceptionV3	512 × 512 × 3 256 × 256 × 3 200 × 200 × 3	42,658,176 14,911,302	82.7
GoogleNet	512 × 512 × 3 256 × 256 × 3 200 × 200 × 3	14,714,688 14,865,222 14,911,302	83.5
Xception	512 × 512 × 3 256 × 256 × 3 200 × 200 × 3	14,714,688 14,865,222 14,911,302	82.4
MobileNet	512 × 512 × 3 256 × 256 × 3 200 × 200 × 3	3,228,864 14,865,222 14,911,302	84.3
SqueezeNet	512 × 512 × 3 256 × 256 × 3 200 × 200 × 3	7,037,504 14,911,302	87.6
**SqueezeNet-Light**	512 × 512 × 3 256 × 256 × 3 200 × 200 × 3	3,182,412 3,182,412 3,182,412	95.6

**Table 8 diagnostics-13-02531-t008:** Results of the proposed system model’s classification using 16 batches of data.

Model	Epochs	SE	SP	ACC	PR	F1-Score	MCC
VGG16	40	78	80	79	76	79	80
AlexNet	40	79	82	81.3	80	80.4	81.1
InceptionV3	40	81	80	82.7	82	82.7	83.5
GoogleNet	40	83	81	83.5	83	83.5	84.5
Xception	40	82	83	82.4	83	84.3	85.4
MobileNet	40	84	84.2	84.3	84	85.2	86.3
SqueezeNet	40	85	86.2	87.6	85	86.1	87.2
Squeeze-Light	40	94	96	95.6	94.12	95.2	96.7

SE: Sensitivity, SP: Specificity, RL: Recall, PR: Precision, ACC: Accuracy.

**Table 9 diagnostics-13-02531-t009:** Results of the proposed system model’s classification using 32 batches of data.

Model	Epochs	SE	SP	ACC	PR	F1-Score	MCC
VGG16	40	78	80	79	76	79	80
AlexNet	40	79	82	81.1	80	80.0	81.0
InceptionV3	40	81	80	82.3	82	82.2	83.4
GoogleNet	40	83	81	83.6	83	83.3	84.3
Xception	40	82	83	82.6	83	84.4	85.2
MobileNet	40	84	84.0	84.3	84	85.1	86.1
SqueezeNet	40	85	86.1	87.2	85	86.0	87.0
Squeeze-Light	40	94	96	95.6	94.12	95.2	96.7

SE: Sensitivity, SP: Specificity, RL: Recall, PR: Precision, ACC: Accuracy.

**Table 10 diagnostics-13-02531-t010:** Results of the proposed system model’s classification using 64 batches of data.

Model	Epochs	SE	SP	ACC	PR	F1-Score	MCC
VGG16	40	78	80	80	76	79	80.5
AlexNet	40	80	81	80.3	80	80.4	82.3
InceptionV3	40	82	82	81.7	82	82.7	84.0
GoogleNet	40	82	83	82.5	83	82.5	85.0
Xception	40	84	84	83.4	83	83.3	86.0
MobileNet	40	83	82.2	85.3	84	84.2	83.0
SqueezeNet	40	85	85.2	86.6	85	85.1	86.1
Squeeze-Light	40	94	96	95.6	94.12	95.2	96.7

SE: Sensitivity, SP: Specificity, RL: Recall, PR: Precision, ACC: Accuracy.

**Table 11 diagnostics-13-02531-t011:** Results of the proposed system model’s classification using various optimizers.

Optimization	SE	SP	ACC	PR	F1-Score	MCC
SGD with Momentum	80.2	81.8	82.9	85	80	82.2
Adam	82	83	82.5	83	82.5	85.0
RMSProp	84	84	83.4	83	83.3	86.0
AdaGrad	83	82.2	85.3	84	84.2	83.0
AdaBelief	94	96	95.6	94.12	95.2	96.7

SE: Sensitivity, SP: Specificity, RL: Recall, PR: Precision, ACC: Accuracy.

**Table 12 diagnostics-13-02531-t012:** Proposed system model’s classification results using various loss functions.

Loss Functions	SE	SP	ACC	PR	F1-Score	MCC
Cross Entropy Loss	82.2	97.8	96.9	76	79	82.2
Weighted Cross Entropy Loss	94	96	95.6	94.12	95.2	96.7

SE: Sensitivity, SP: Specificity, RL: Recall, PR: Precision, ACC: Accuracy.

**Table 13 diagnostics-13-02531-t013:** Computational performance of different architectures.

DL Architectures	Complexity (MFLOPs)	Model Size (MB)	GPU Speed (S)
SqueezeNet-Light	68.3	9.3	0.7
SqueezeNet	96.9	14.5	1.6
MobileNet	95.4	12.3	1.3
GoogleNet	272.8	15.2	2.7
Xception	281.8	16.3	2.6
InceptionV3	554.3	17.5	3.0
AlexNet	65.9	14.5	2.8
VGG16	295.8	12.3	3.4

MFLOPS: million floating-point operations per second, M: millions, MB: megabytes, S: seconds.

**Table 14 diagnostics-13-02531-t014:** Performance of CPU/TPU/GPU Comparisons of the proposed SqueezeNet-Light model.

Batch Size	Epochs	CPU/TPU/GPU (mS)
64	40	700/500/400
128	40	750/400/500
256	40	750/400/500
512	40	750/400/500
1024	40	750/400/500

mS: milliseconds, CPU: central processing unit, GPU: graphical processing unit, and TPU: tensor processing units.

**Table 15 diagnostics-13-02531-t015:** Classification results of the SOTA systems.

Model	Epochs	SE	SP	ACC	PR	F1-Score	Trainable Parameters
SqueezeNet-Light	100	94	96	95.6	94.12	95.2	3,182,412
SqueezeNet	100	87	85	87.6	87	87	7,037,504
Salama-ResNet-SVM [10]	100	80	82	81	79	80	53,982,272
Ashraf-FRCNN [15]	100	82	83	82	80	81	50,213,111
Naeem-VGG16 [26]	100	83	85	83	83	84	48,222,341
Hosny-AlexNet [34]	100	84	86	84	84	85	49,112,242
Fujisawa-DCNN [37]	100	79	80	78	79	80	52,128,141
Harangi-GoogleNet-Inception [39]	100	85	86	85	84	85	50,440,122

SE: sensitivity, SP: specificity, PR: precision, ACC: accuracy.

**Table 16 diagnostics-13-02531-t016:** Computational performance of different SOTA architectures.

SOTA Architectures	Complexity (MFLOPs)	Parameters (M)	Model Size (MB)	GPU Speed (S)
SqueezeNet-Light	68.3	20.18	9.3	0.7
SqueezeNet	96.9	38.11	14.5	2.6
Salama-ResNet-SVM [10]	120.3	53.98	22.3	2.7
Ashraf-FRCNN [15]	140.9	50.21	20.5	5.6
Naeem-VGG16 [26]	150.3	48.22	19.3	4.7
Hosny-AlexNet [34]	122.3	49.11	17.3	4.7
Fujisawa-DCNN [37]	120.9	52.13	16.5	3.6
Harangi-GoogleNet-Inception [39]	150.3	50.44	19.3	3.7

MFLOPS: Million floating-point operations per second, M: millions, MB: megabytes, S: seconds.

**Table 17 diagnostics-13-02531-t017:** Limitations and advantages of different SOTA architectures compared to proposed architecture.

TL Architectures	Limitations	Advantages
SqueezeNet-Light	It should be trained on more different datasets and this classifier can be tested on different modality of images to check the generalizability of the model.	Tiny model, high speed, several datasets are evaluated, and identifies nine classes
SqueezeNet	It is small model but requires hyper-parameter tuning	It is better classifier compared to other pretrained TL models.
Salama-ResNet-SVM [10]	Classes imbalance, binary classification only two classes, and computationally expensive.	Integration of SVM and ResNet and good for binary decision.
Ashraf-FRCNN [15]	Classes imbalance, evaluated on single dataset, classify only two classes, and computationally expensive.	This approach used CNN and is better for features extraction.
Naeem-VGG16 [26]	Four classes only, many hyper-parameters required, and tested on signal dataset so classifier is not generalized.	This method used pre-trained TL VG-16 to recognize four classes of PSLs
Hosny-AlexNet [34]	Three classes only, required huge parameters, evaluate on single dataset, and computationally expensive.	This approach used CNN and is better for features extraction.
Fujisawa-DCNN [37]	Two classes (benign vs. malignant), required huge parameters, evaluate on single dataset, and computationally expensive.	This approach used CNN and is better for features extraction.
Harangi-GoogleNet-Inception [39]	Seven classes, required huge parameters, evaluate on single dataset, and computationally expensive.	Combining the GoogleNet and Inception pretrained TL to recognize nine classes.

## Data Availability

Datasets used in these experiments are available online. Following are links used to access these datasets such as: [32,33] HAM10000 https://www.kaggle.com/datasets/kmader/skin-cancer-mnist-ham10000 [Accessed on 12 January 2023]; [42] ISIC 2019: https://challenge.isic-archive.com/landing/2019/ [Accessed on 12 January 2023]; [43] ISIC 2020: https://challenge2020.isic-archive.com/ [Accessed on 12 January 2023]; [44] Ph2: https://www.kaggle.com/datasets/synked/ph2-modified [Accessed on 12 January 2023]; [45] ISBI-2017 https://skinclass.de/TestSet.zip [Accessed on 12 January 2023].

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
