# Peer review of "Assist-Dermo: A Lightweight Separable Vision Transformer Model for Multiclass Skin Lesion Classification"

_diagnostics, 2023, doi:10.3390/diagnostics13152531_

Round 1

Reviewer 1 Report

The manuscript presents a new approach for the automatic classification of pigmented skin lesions (PSLs) using a separable vision transformer (SVT) technique. The manuscript addresses the challenge of high computational requirements in existing systems and aims to develop a lightweight model with competitive accuracy. The proposed Assist-Dermo system incorporates data augmentation, pre-processing techniques, and a modified SqueezeNet design with depthwise separable convolutions. The experimental results demonstrate the efficacy of Assist-Dermo in accurately classifying PSLs, outperforming state-of-the-art algorithms. The manuscript is well-structured and provides clear explanations of the proposed approach, dataset, and evaluation metrics.

The introduction provides a good overview of the problem and the significance of automatic classification systems for skin lesions. However, it would be helpful to present more recent work on lightweight CNN models for automatic classification, such as: doi.org/10.3390/diagnostics13061100; doi.org/10.1007/s00170-022-10335-8.

In the methodology section, consider providing more details on the SVT architecture based on SqueezeNet and depthwise-separable CNN models. Clarify how these models are combined and modified to create the Assist-Dermo system. Providing a visual representation or a diagram of the SVT architecture would aid in understanding its structure and components. While the manuscript mentions that the Assist-Dermo system outperformed state-of-the-art algorithms, it would be beneficial to include a comparison table or additional information on the performance of other competitive systems. This would allow readers to better understand the relative improvements achieved by Assist-Dermo.

In the conclusion, it is suggested to highlight how the lightweight nature of the model makes it suitable for deployment on resource-constrained devices, enabling wider access to accurate skin lesion classification.

The language and style of the manuscript are generally clear and concise, making it easy to understand the proposed approach and experimental results.

Author Response

Original Manuscript ID:  ID: diagnostics-2493117       

Original Article Title: Assist-Dermo: A Lightweight Separable Vision Transformer Model for Multiclass Skin Lesion Classification

To: Editor in Chief,

MDPI, Diagnostics

Re: Response to reviewers

Dear Editor,

Many thanks for insightful comments and suggestions of the referees. Thank you for allowing a resubmission of our manuscript, with an opportunity to address the reviewers’ comments.

We are uploading (a) our point-by-point response to the comments (below) (response to reviewers), (b) an updated manuscript with yellow highlighting indicating changes, and (c) a clean updated manuscript without highlights (PDF main document).

By following reviewers’ comments, we made substantial modifications in our paper to improve its clarity, English and readability. In our revised paper, we represent the improved manuscript such as:

(1) Revised Abstract, (2) Revised Introduction, (3) Results section, (4) Discussions and Conclusion sections.

We have made the following modifications as desired by the reviewers:

Best regards,

Corresponding Author,

Dr. Qaisar Abbas (On behalf of authors),

Professor.

Reviewer 1:

Comment - (1) The introduction provides a good overview of the problem and the significance of automatic classification systems for skin lesions. However, it would be helpful to present more recent work on lightweight CNN models for automatic classification, such as: doi.org/10.3390/diagnostics13061100; doi.org/10.1007/s00170-022-10335-8.

Response 1: As, You are right. As advised by you, we have added these two papers and explained them in the discussion and conclusion section. We have added the following references as:

  1. Zhu, R.; Cui, Y.; Huang, J.; Hou, E.; Zhao, J.; Zhou, Z.; Li, H. YOLOv5s-SA: Light-Weighted and Improved YOLOv5s for Sperm Detection. Diagnostics 2023, 13, 1100. [CrossRef]
  2. Li, W., Zhang, L., Wu, C., Cui, Z., & Niu, C. (2022). A new lightweight deep neural network for surface scratch detection. The International Journal of Advanced Manufacturing Technology, 123(5-6), 1999-2015. [CrossRef]

In the discussion, we have added as:

In the future, we will also compare this proposed SqueezeNet-Light architecture with other recently developed lightweight [49, 50] architectures to confirm the generalizability of the network. 

In the conclusion section, we have added as:

Computational efficiency shows that the proposed system can be easily deployed in an environment where is a requirement of resource-constraint devices such as mobile devices. However, it is important to note that the lightweight nature of the model should also be tested on the Internet of Things (IoT) devices, which makes it suitable for deployment on a wider scale for the accurate classification of pigmented skin lesions (PSLs). This point-of-view will be addressed in future works.

Those changes can be easily seen in the revised paper. Thank you to clear this problem in our paper.

Comment - (2) In the methodology section, consider providing more details on the SVT architecture based on SqueezeNet and depthwise-separable CNN models. Clarify how these models are combined and modified to create the Assist-Dermo system. Providing a visual representation or a diagram of the SVT architecture would aid in understanding its structure and components. While the manuscript mentions that the Assist-Dermo system outperformed state-of-the-art algorithms, it would be beneficial to include a comparison table or additional information on the performance of other competitive systems. This would allow readers to better understand the relative improvements achieved by Assist-Dermo.

Response 2: As suggested by reviewer #1, we have add more emphasis on the medical side of ophthalmological data compared to technical details. Those changes can be easily seen on page# 2 by adding a new table, which is table 1.

So, we have done the following modifications with respect to this comment.

Thank you to clear this problem in our paper.

Comment - (3) In the conclusion, it is suggested to highlight how the lightweight nature of the model makes it suitable for deployment on resource-constrained devices, enabling wider access to accurate skin lesion classification.

Response 3: Yes, we have done this change as suggested by you.  The following paragraph is added in the conclusion section to explain this point.

Computational efficiency shows that the proposed system can be easily deployed in an environment where is a requirement of resource-constraint devices such as mobile devices. However, it is important to note that the lightweight nature of the model should also be tested on the Internet of Things (IoT) devices, which makes it suitable for deployment on a wider scale for the accurate classification of pigmented skin lesions (PSLs). This point-of-view will be addressed in future works.

Thank you for this valuable comment to increase the readability of the manuscript.

Reviewer 2 Report

1. The authors are suggested to modify the opening sentence of the abstract. They can write something like,” This paper introduces a novel automated classification system, inspired by dermatologists, which utilizes a separable vision transformer (SVT) technique. The system aims to identify and categorize nine distinct classes of pigmented skin lesions (PSLs). Its purpose is to support clinical experts in the early detection of skin cancer.” 

2. The authors are suggested to reverify the numbers about cancer given in the introduction. 

3. As already authors have given a table for recent literature, they can reduce this content in the text to improve the readability.

4. There are many grammatical corrections required in the entire article, hence refine the English language.

5. The methodology section is interesting, but a better presentation is required.

The authors need to refine the English Language. 

Author Response

Original Manuscript ID:  ID: diagnostics-2493117       

Original Article Title: Assist-Dermo: A Lightweight Separable Vision Transformer Model for Multiclass Skin Lesion Classification

To: Editor in Chief,

MDPI, Diagnostics

Re: Response to reviewers

Dear Editor,

Many thanks for insightful comments and suggestions of the referees. Thank you for allowing a resubmission of our manuscript, with an opportunity to address the reviewers’ comments.

We are uploading (a) our point-by-point response to the comments (below) (response to reviewers), (b) an updated manuscript with yellow highlighting indicating changes, and (c) a clean updated manuscript without highlights (PDF main document).

By following reviewers’ comments, we made substantial modifications in our paper to improve its clarity, English and readability. In our revised paper, we represent the improved manuscript such as:

(1) Revised Abstract, (2) Revised Introduction, (3) Results section, (4) Discussions and Conclusion sections.

We have made the following modifications as desired by the reviewers:

Best regards,

Corresponding Author,

Dr. Qaisar Abbas (On behalf of authors),

Professor.

Reviewer #2:

Comment - (1 The authors are suggested to modify the opening sentence of the abstract. They can write something like,” This paper introduces a novel automated classification system, inspired by dermatologists, which utilizes a separable vision transformer (SVT) technique. The system aims to identify and categorize nine distinct classes of pigmented skin lesions (PSLs). Its purpose is to support clinical experts in the early detection of skin cancer.”

Response 1: As suggested by reviewer #2, we have updated all those sentences in the Abstract part, which are having some unclear sentences. Also, we have adjusted grammatical mistakes and typos errors through the paper. Yes, you are right, there were English writing problems in the first version of the paper but, it has been improved in terms of writing. We have used professional software to improve the writing. We have carefully read the whole paper and improve it. You can find these changes in the revised paper through word tracking.

New changes can be found in the Abstract part as:

A Dermatologist-like automatic classification system is developed in this paper to recognize nine different classes of pigmented skin lesions (PSLs) using a separable vision transformer (SVT) technique to assist clinical experts in early skin cancer detection. In the past, researchers have developed a few systems to recognize nine classes of PSLs. However, they often require enor-mous computations to achieve high performance, which is burdensome to deploy on re-source-constrained devices. In this paper, a new approach to designing the SVT architecture is developed based on SqueezeNet and depthwise-separable CNN models. The primary goal is to find a deep learning architecture with few parameters that has comparable accuracy to state-of-the-art (SOTA) architectures. This paper modifies the SqueezeNet design for improved runtime performance by utilizing depthwise separable convolutions rather than simple conven-tional units. To develop this Assist-Dermo system, a data augmentation technique is applied to control the PSL imbalance problem. Next, a pre-processing step is integrated to select the most dominant region and then enhance the lesion patterns in a perceptual-oriented color space. Af-terwards, the Assist-Dermo system is designed to improve efficacy and performance with several layers and multiple filter sizes but fewer filters and parameters. For the training and evaluation of Assist-Dermo models, a set of PSL images is collected from different online data sources such as Ph2, ISBI-2017, HAM10000, and ISIC to recognize nine classes of PSLs. On the chosen dataset, it achieves an accuracy (ACC) of 95.6%, a sensitivity (SE) of 96.7%, a specificity (SP) of 95%, and an area under the curve (AUC) of 0.95. The experimental results show that the suggested As-sist-Dermo technique outperformed SOTA algorithms when recognizing nine classes of PSLs. The Assist-Dermo system performed better than other competitive systems and can support derma-tologists in the diagnosis of a wide variety of PSLs through dermoscopy. The Assist-Dermo model code is freely available on GitHub for the scientific community.. 

Thank you to clear this problem in our paper.

Comment - (2) The authors are suggested to reverify the numbers about cancer given in the introduction.?

Response 2: Yes, you are right. In the revised version of the paper, we have updated the cancer statistical data.

Now in the updated paragraph of introduction as:

Skin cancer is becoming more widespread in the Western world, with significant ramifications for both general skincare and the availability of dermatological treatments. Day after day, about 99,780 individuals in the United States are identified with melanoma or skin cancer. Among them, two or more are likely to die per hour. Skin cancer affects more individuals in the United States each year compared to other cancers combined [1]. Europe accounts for 9% of the global population yet bears 25% of the worldwide cancer cases. If tumors are recognized and treated early, cancer mortality can be considerably decreased. Thus, it is crucial to devote research resources to implementing systems for primary cancer recognition.

Thank you to point out this viewpoint.

Comment - (3) As already authors have given a table for recent literature, they can reduce this content in the text to improve the readability.?

Response 3: Yes, you are right. In the revised version of the paper, we have decreased the length of literature review. However, there are important information which we do not want to update.

These changes can be easily seen in page# 3 and start from line# 106.

Thank you for these valuable comments.

Comment - (4) There are many grammatical corrections required in the entire article, hence refine the English language.

Response 4: As suggested by reviewer #2, we have updated all those sentences in the Abstract, Introduction, Literature review and all other parts, which have some unclear sentences. Also, we have adjusted grammatical mistakes and typos errors through the paper. Yes, you are right, there were English writing problems in the first version of the paper but, it has been improved in terms of writing. We have used professional software to improve the writing. We have carefully read the whole paper and improved it. You can find these changes in the revised paper through word tracking.

Comment - (5) The methodology section is interesting, but a better presentation is required.

Response 5: As suggested by reviewer #2, It is impossible to give as better explanation as presented in the revised version. Algorithm 1 and figures of proposed methodology are very clear. The proposed steps are presented in an algorithm form.

Thank you for the comment.
